# Co-transmission of neuropeptides and monoamines choreograph the *C. elegans* escape response

**Jeremy T. Florman**, **Mark J. Alkema** *

Department of Neurobiology, UMass Chan Medical School, Worcester, Massachusetts, United States of America

* mark.alkema@umassmed.edu

## Abstract

Co-localization and co-transmission of neurotransmitters and neuropeptides is a core property of neural signaling across species. While co-transmission can increase the flexibility of cellular communication, understanding the functional impact on neural dynamics and behavior remains a major challenge. Here we examine the role of neuropeptide/monoamine co-transmission in the orchestration of the *C. elegans* escape response. The tyraminergic RIM neurons, which coordinate distinct motor programs of the escape response, also co-express the neuropeptide encoding gene *flp-18*. We find that in response to a mechanical stimulus, *flp-18* mutants have defects in locomotory arousal and head bending that facilitate the omega turn. We show that the induction of the escape response leads to the release of FLP-18 neuropeptides. FLP-18 modulates the escape response through the activation of the G-protein coupled receptor NPR-5. FLP-18 increases intracellular calcium levels in neck and body wall muscles to promote body bending. Our results show that FLP-18 and tyramine act in different tissues in both a complementary and antagonistic manner to control distinct motor programs during different phases of the *C. elegans* flight response. Our study reveals basic principles by which co-transmission of monoamines and neuropeptides orchestrate in arousal and behavior in response to stress.

## Author summary

Co-transmission is a form of neuronal communication where multiple signaling molecules are released from an individual cell. It is commonly found in nervous systems across species that monoamines and neuropeptides are co-released and affect the properties of target cells. In humans, adrenaline and neuropeptide Y are co-released from sympathetic neurons where they increase blood pressure and heart rate during the fight-or-flight response. The complexity of the human nervous system makes it very hard to figure out how these signaling molecules interact to change behavior and physiology. Here, we use the nematode *C. elegans* to study how co-transmission of the invertebrate analog of adrenaline, tyramine, and a neuropeptide, FLP-18, coordinate the distinct motor programs of the flight response. We find that FLP-18 and tyramine act together to shape different

**Data Availability Statement:** All relevant data are within the manuscript and its Supporting Information files.

**Funding:** This work was supported by NIH grant GM140480 and NS107475 from the National

Institutes of Health (MJA). The funders had no role in study design, data collection and analysis, decision to publish, or preparation of the manuscript.

**Competing interests:** The authors have declared that no competing interests exist.

phases of the flight response. Our findings illuminate the cellular mechanisms by which co-transmission of monoamines and neuropeptides orchestrate a flight response.

## Introduction

Co-localization of classical neurotransmitters and neuropeptides is a common feature of the animal nervous system [1]. Co-transmission is thought to provide flexibility to the output of hard-wired neural circuits [2]. In mammals, for instance, the acute fight-or-flight response leads to the activation of the sympathetic nervous system (SNS) and the co-release of different "stress hormones" [3]. Adrenalin and noradrenaline release in the SNS trigger an increase in heart rate, blood flow, respiration and release of glucose from energy stores, which prepare the animal for vigorous muscle activity and physical exertion [4,5]. Neuropeptide Y (NPY), one of the most abundant neuropeptides in the mammalian nervous system, is a co-transmitter with noradrenaline (NA) in many neurons of the SNS [6–8]. The sympathetic co-transmission of NA and NPY suggest that they may coordinate aspects of the flight response. However, the physiological and behavioral impact of co-transmission can be difficult to dissect. Co-transmitted signaling molecules can activate receptors on a common target (convergence) or different targets (divergence) which induce synergistic or opposing effects, making the functional outcome of co-transmission challenging to anticipate. Studies in mammals have shown that the stress related modulatory functions of NPY and NA co-transmission are complex, with complementary actions in some tissues and antagonistic actions in others. For example, both NA and NPY increase blood pressure through peripheral vasoconstriction, however NPY inhibits presynaptic NA release from sympathetic neurons and opposes the action of NA on cardiac contraction [9–19]. Co-transmission of NPY and catecholamines may induce a longer lasting state of arousal that enhances alertness and the ability to deal with environmental threats. Unraveling the precise effects of co-transmission of neural stress hormones is very challenging in vertebrates given the complexity of the nervous system, the multiple central and peripheral release sites and the diversity of target tissues expressing NA and NPY receptor (NPYR) subtypes.

The nematode *Caenorhabditis elegans* provides an excellent system to study the co-transmission of aminergic and peptidergic neuromodulators due to its compact and completely defined nervous system and wealth of genetic tools [20,21]. The *C. elegans* genome encodes a large family of NPY related peptides and G-protein coupled receptors (GPCRs) [22,23]. Like other invertebrates, *C. elegans* lacks NA, however the structurally related tyramine fulfills a similar role to NA in coordinating stress responses and flight behavior [24–26].

In response to a mechanical stimulus *C. elegans* can engage in a flight- or an escape-response, where the worm quickly reverses while suppressing oscillatory head movements. The reversal is followed by a deep ventral bend of the head, and a subsequent slide of the head along the ventral side of the body (omega turn). After the omega turn, the animal moves forward in the opposite direction, away from the noxious stimulus. Tyramine plays a crucial role in the coordination of independent motor programs, which increases the animal's chances to escape from predatory fungi that use hyphal nooses to entrap nematodes [26–29]. The escape response triggers the release of tyramine from a single pair of neurons called the RIM. Tyramine release activates the inhibitory tyramine-gated chloride channel, LGC-55, in neck muscles, cholinergic head motor neurons and the AVB pre-motor interneuron. LGC-55 activation induces the suppression of oscillatory head movements during long reversals in response to anterior touch [26,28,30,31]. In addition, tyramine release facilitates ventral bending during

the omega turn through the activation of the SER-2 GPCR in GABAergic motor-neurons [29]. The tyraminergic RIM neurons co-express a NPY like peptide, FLP-18 [32,33]. The RFamide FLP-18 and its associated receptors are related to the NPY/NPYR signaling system [34]. *In vitro* experiments have shown that FLP-18 can activate a human NPYR and, conversely, human NPY can activate worm neuropeptide receptors [35]. In *C. elegans*, FLP-18 has been shown to play a role in foraging and metabolism [36], arousal and homeostasis [37–39], chemosensation [40], reversal behavior [41,42], and swimming [33,43]. FLP-18 acts through several neuropeptide receptors including NPR-1, NPR-4, and NPR-5 [32,36,44].

Here we investigate the role of FLP-18 in the orchestration of the *C. elegans* escape response. We find that FLP-18 plays a central role in coordinating distinct phases of the escape response in *C. elegans*. FLP-18 is co-released with tyramine from the RIM neurons in response to mechanical stimuli. FLP-18 activates the GPCR NPR-5 in body wall muscle to enhance turning behavior and locomotion speed during the escape response by increasing muscle excitability. Our result show that FLP-18 and tyramine act in different tissues in both a complementary and antagonistic manner to orchestrate distinct motor programs during different phases of the *C. elegans* flight response.

## Results

### Mechanical stimulation transiently increases forward velocity

Handling of *C. elegans*, such as the transfer with a platinum wire, bumping the plate on microscope stage or removing the lid of a plate can induce an increase in locomotion rate that persists for several minutes [45,46]. This indicates that mechanical stimulation can lead to longer lasting changes to the internal state of the animal. To quantify locomotion patterns upon mechanical stimulation we analyzed the locomotion rate of animals subjected to a tap to the side of the plate. Mechanical tap can trigger an escape response, in which *C. elegans* reverses, turns and resumes forward locomotion in the opposite direction [47,48]. We compared behavior of animals before, during and after spontaneous reversals and tap-induced reversals. In animals that initiated a spontaneous reversal, forward locomotion rate remained the same before and after the reversal (Fig 1A). The absolute velocity during spontaneous reversals was similar to the velocity of forward movement.

When an escape response was triggered with a tap stimulus, animals initiated a rapid reversal; double the speed of spontaneous reversals (Fig 1B and 1C). Furthermore, following a tap-induced reversal, wild-type animals exhibited a markedly elevated forward locomotion rate for approximately 3 minutes (S1A Fig);—a behavior we refer to as the 'forward run' (Fig 1B and 1D). The sustained increase in forward velocity in response to a tap indicated longer lasting changes in the internal state of the animal.

### *flp-18* mutants have defects in locomotory arousal

Neuropeptides are potent neuromodulators, whose action can lead to longer lasting changes in behavior. We focused our attention on FLP-18 neuropeptides, since the *flp-18* gene is known to be expressed in the tyraminergic RIM neurons and the AVA pre-motor interneurons [32,33]. The RIM and AVA play a central role in the escape response to mechanical touch [25,26,47,49]. *flp-18* mutants and tyramine deficient *tdc-1* mutants initiated reversals in response to tap stimuli similar to wild type (Fig 1E). This indicates that FLP-18 peptides and tyramine are not required for mechanosensation or the initiation of the escape response. In response to a tap stimulus, *flp-18* mutants reversal velocity was slightly reduced compared to wild type and *tdc-1* mutant animals (Fig 1B and 1C). Both *flp-18* and *tdc-1* mutants exhibited shorter reversal duration compared to wild type (Fig 1B). *tdc-1* mutants had a reduced basal

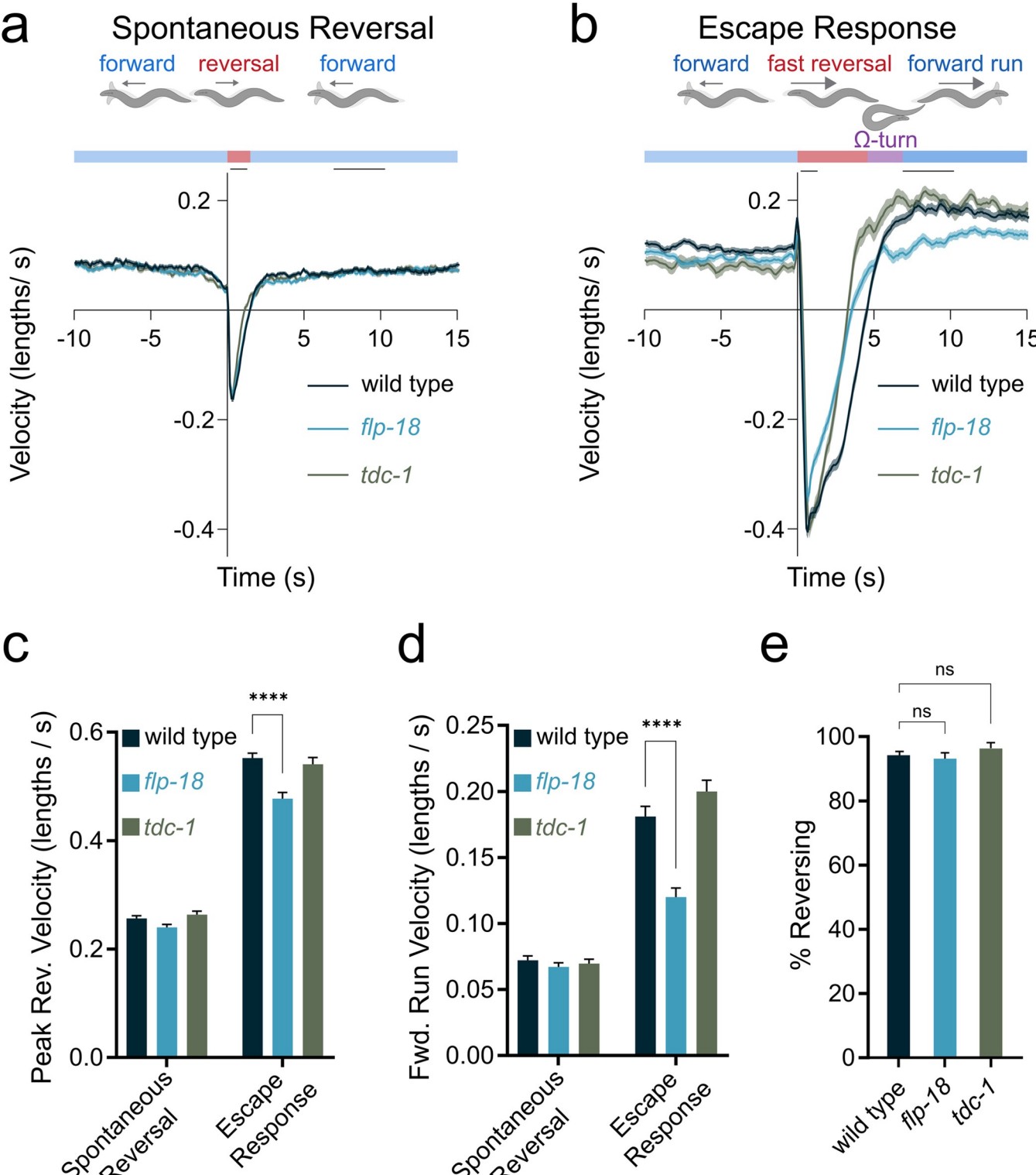

**Fig 1. Increase in locomotion speed following a mechanical stimulus.** Backward- and forward-velocity upon a spontaneous or tap-induced reversal. Wild-type animals show a persistent increase in locomotion speed in response to a strong tap stimulus ("forward run"). *flp-18* mutants do not show a robust increase locomotion speed in response to a strong tap stimulus. (A-B) Schematic illustrating behavioral sequence (top) and velocity traces (bottom) from wild type animals, *flp-18*, and *tdc-1* mutants, aligned to reversal events. Negative velocity indicates backward locomotion. Dark lines represent mean velocity and shaded region represents standard error. Black lines at the top of the graph indicate the time periods when reversal and forward run speed were quantified. Reversals

were spontaneous (A) or induced by a tap stimulus (B). (C) Peak backward velocity during the reversal phase was quantified by identifying the maximum backward locomotion rate for each animal during a 1 second period following the tap. (D) Forward run velocity was quantified as the average speed for each animal during the time period between 7 to 10 seconds following the tap. (E) Quantification of the percentage of animals initiating a reversal within 1 second after a tap was delivered. Graphs represent mean ± SEM, significance was calculated using ANOVA with Šidák's (C and D) and Dunnett's (E) multiple comparison test (P> 0.05 = ns, P<0.005 = **, P<0.0001 = ****). Sample sizes: Spontaneous reversals and escape response (A-D), (n = # animals). Spontaneous (A, C, and D): wild type (n = 506), *flp-18* (n = 512), *tdc-1* (n = 512). Escape (b-d): wild type (n = 275), *flp-18* (n = 208), *tdc-1* (n = 124). % Reversing (E), (n = # of experiments, 20 worms per experiment): wild type (n = 30), *flp-18* (n = 23), *tdc-1* (n = 11).

velocity, but following a tap, increased velocity during the forward run, similar to the wild type. In contrast, *flp-18* mutants had significantly slower forward run velocity compared to the wild type (Fig 1B and 1D). *tdc-1; flp-18* double mutants behaved similar to *tdc-1* single mutants during the initial phase of the escape response and like *flp-18* single mutants failed to maintain forward velocity at later stages of the forward run (S1B Fig).

## *flp-18* mutants have defects in head and body bending during the escape response

Under regular conditions *flp-18* mutants had no obvious defects in locomotion velocity, body curvature, foraging head movements, or spontaneous reversal frequency (Fig 2A–2F) and propagated regular body bends as they moved across the agar surface (Fig 2G). We analyzed *flp-18* mutant animals for defects in other aspects of the escape response, since *flp-18* mutants fail to sustain rapid movement during the forward run. During the initial phase of the escape response elicited by gentle anterior touch, animals reverse while suppressing oscillatory head movements. The reversal is often followed by two linked motor programs that together compose an omega turn: a ventral head swing which initiates the omega turn; followed by a deep body bend where the animal slides its head along the ventral side of the body to resume locomotion in the opposite direction. In response to gentle anterior touch *flp-18* mutants reversed and suppressed oscillatory head movements like the wild type (Fig 3E). Following a reversal, *flp-18* mutants often initiate an omega turn similar to the wild type. However, *flp-18* mutants made shallower ventral head bends (Fig 3B) and executed omega turns (> 90-degree turn) less frequently than wild-type animals (37% vs 57% respectively) (Fig 3C). Furthermore, when *flp-18* mutants did make high-angled omega turns, the head often failed to contact the ventral side of the body during the turn [29] (Fig 3A) resulting in a much higher proportion of open omega turns compared to wild-type animals (62% vs 22% respectively) (Fig 3D). Expression of a low-copy transgenic *flp-18* genomic construct, which includes a 3 kb endogenous promoter ([P*flp-18*::FLP-18], referred to as FLP-18(+)), rescued the omega turning defects seen in *flp-18* mutants (Fig 3C–3E).

## *flp-18* overexpression enhances body bending and omega turning

To further assess the effect of FLP-18 signaling on behavior we overexpressed *flp-18* using a high-copy transgene ([P*flp-18*::FLP-18], referred to as FLP-18(+++)). *flp-18* overexpression caused uncoordinated locomotion with strikingly deep body bends (Fig 2A and 2G) consistent with previous reports [32]. Animals overexpressing *flp-18* also displayed hyperactive foraging head movements, increased body curvature, and increased spontaneous reversal frequency (Fig 2C–2F). *flp-18* reporters are expressed in the AVA, RIM, AIY, RIG, the M1 and M3 pharyngeal neurons and ventral cord motor neurons [32,33,39]. We overexpressed *flp-18* in a subset of neurons of the escape circuit using the *cex-1* promoter, which mainly drives expression in the RIM, AVA and AVD neurons [50], but not the AIY, RIG, pharyngeal and ventral cord neurons (S2A Fig). Overexpression of the P*cex-1*::FLP-18 transgene in a *flp-18* mutant background was sufficient to cause exaggerated body bends and reversals, similar to P*flp-18*::FLP-

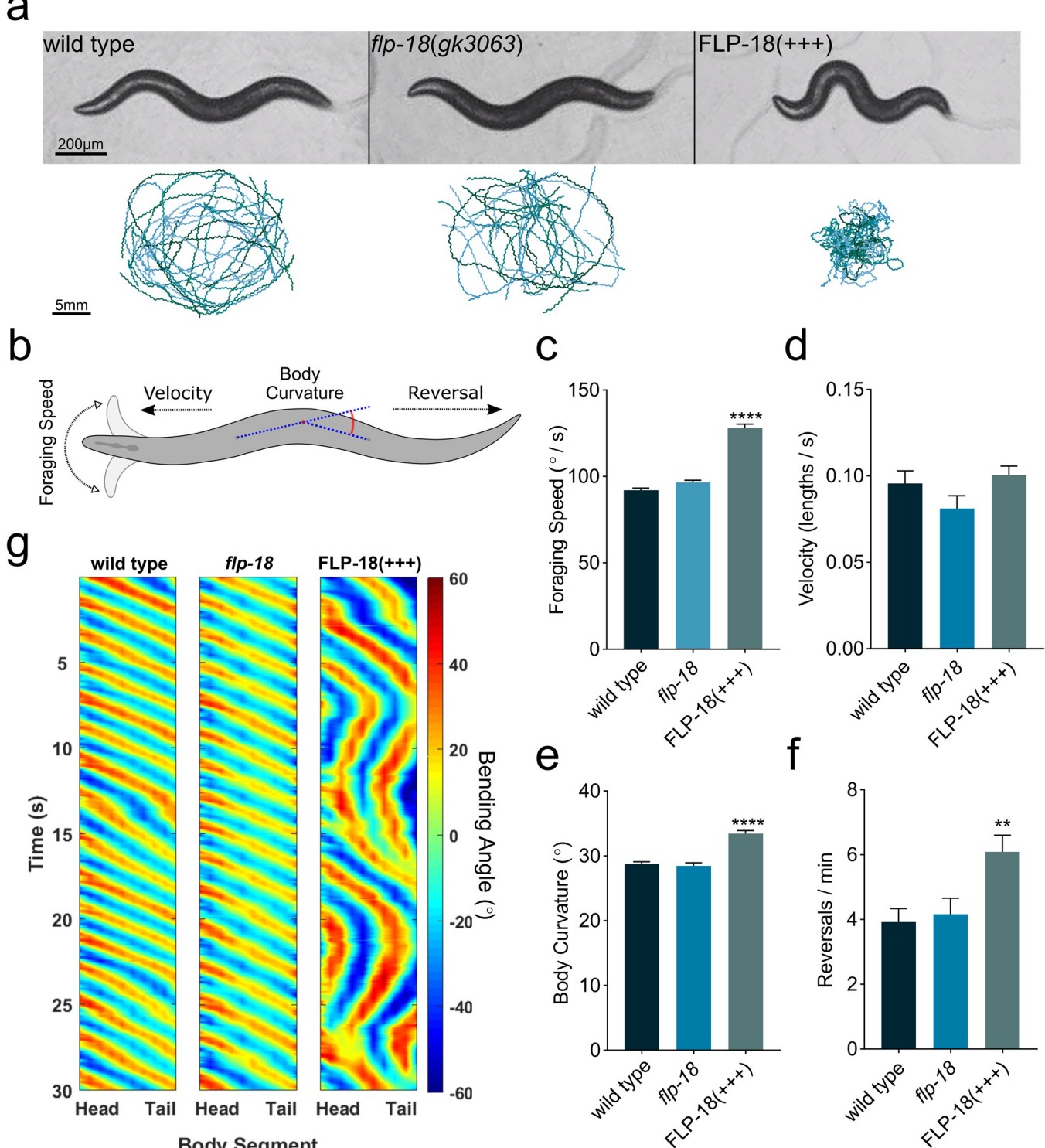

**Fig 2. Overexpression of FLP-18 causes locomotion defects.** (A) Behavioral tracking of wild type, *flp-18(gk3063)* mutants, and transgenic animals that overexpress FLP-18 (FLP-18(+++)). Representative images of each genotype (upper panel) and locomotion tracks recorded from multiple animals freely moving on a plate for 5 min (lower panel). Scale bars represent 200 μm (top) and 5 mm (bottom). (B) Schematic illustrating quantified behaviors. (C-F) Quantification of locomotion behavior. (C) Speed of foraging head movements. (D) locomotion velocity. (E) Body bend curvature. (F) Spontaneous reversal frequency. (G) Kymographs showing body bending along the body over 30 s of locomotion. Color map represents degrees of bending, positive values represent dorsal bends and negative values represent ventral bends. Graphs represent mean ± SEM, significance was calculated using ANOVA with Dunnett's multiple

comparison test (P<0.005 = \*\*, P<0.0001 = \*\*\*\*). Sample sizes: (C), wild type (n = 38), *flp-18* (n = 28), FLP-18(+++) (n = 31). (D-F) wild type, *flp-18*, and FLP-18(+++) (n = 16 experiments per genotype, 20 animals per experiment).

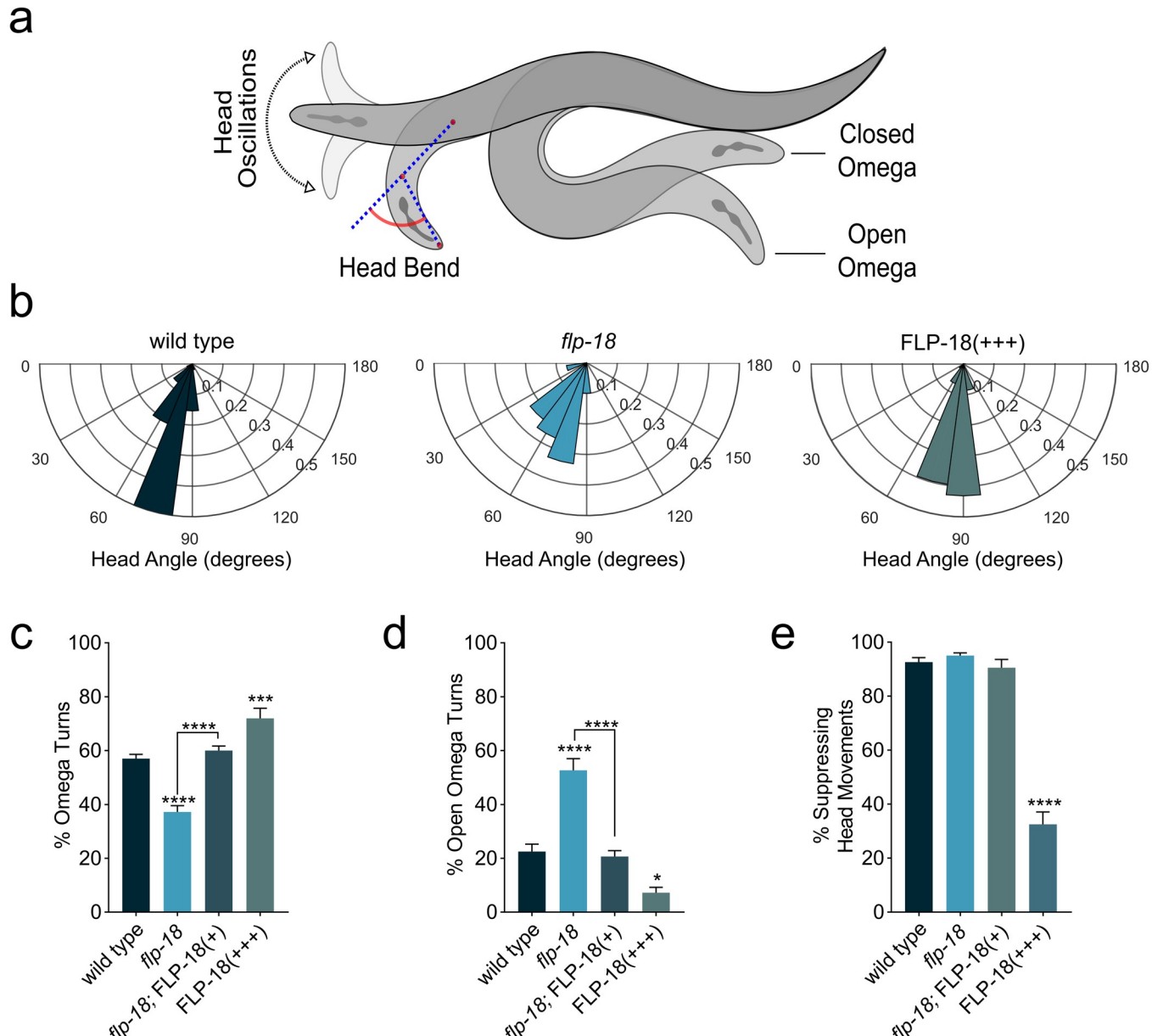

**Fig 3. *flp-18* mutants are defective in head bending and turning during the escape response.** (A) Schematic depicting head oscillations, ventral turn head angle and open vs closed omega turns. (B) Angle of ventral head bend following touch induced reversal of animals that make omega turns (probability histogram with a bin size of 15˚). Each concentric circle represents a probability of 0.1. (C-D) Quantification of omega turning behavior. (C) Percentage of animals that execute omega turns in response to gentle anterior touch. (D) Fraction of open omega turns out of total omega turns. (E) Quantification of the percentage of animals suppressing head oscillations upon gentle anterior touch. Mean ± SEM. significance was calculated using ANOVA with Šidák's multiple comparison correction (P<0.05 = \*, P<0.0005 = \*\*\*, P<0.0001 = \*\*\*\*). Sample sizes: Ventral head bend measurement (B) (n = # of animals), wild type (n = 52), *flp-18* (n = 30), FLP-18(+++) (n = 58). Suppression of head movement and omega quantification (C-E) (n = # of experiments, 20 worms per experiment) wild type (n = 17), *flp-18* (n = 16), *flp-18;* FLP-18(+) (n = 9), FLP-18(+++) (n = 10).

18 transgenic animals (S2B–S2D Fig and S1 and S2 Videos). This indicates that FLP-18 release from the escape circuit neurons, AVA, RIM and AVD, is sufficient to enhance body bending and reversals.

FLP-18 overexpression had an opposite effect on the escape response compared to *flp-18* deficiency. In contrast to *flp-18* mutants, which have reduced ventral head bend angle, FLP-18 overexpressing animals made high angled head bends (Fig 3B). Furthermore, overexpression of FLP-18 resulted in an increased omega turn frequency and a reduced proportion of open omega turns (Fig 3C and 3D). Moreover, in response to a mechanical stimulus FLP-18 overexpressing animals failed to suppress head movements during reversals (Figs 3E and S3 and S3 and S4 videos). This phenotype is similar to tyramine deficient mutants (S1C Fig) and mutants that lack a tyramine-gated chloride channel, LGC-55 [26,28,30]. In wild-type animals, suppression of head oscillations during touch induced reversals is mediated by tyraminergic activation of LGC-55 in neck muscles and SMD and RMD head motor-neurons. The observation that animals overexpressing FLP-18 fail to suppress head movements suggests that FLP-18 may oppose the tyraminergic inhibition of neck muscles.

### The escape response induces FLP-18 release from the AVA and RIM

*flp-18* is expressed in the AVA and RIM neurons [32,33], which are activated during the escape response [25,49,51]. To analyze FLP-18 release, we generated transgenic animals with a fluorescently tagged the FLP-18 pro-peptide with the YFP variant Venus (P*flp-18*::FLP-18::Venus). Venus is resistant to quenching in low pH environments and has been used to monitor neuropeptide secretion from dense core vesicles in *C. elegans* [52–54]. FLP-18::Venus fluorescence was observed in the RIM and AVA escape circuit neurons, as well as the AIY and RIG (Fig 4B). FLP-18::Venus fluorescence was also observed in the coelomocytes, in contrast to a P*flp-18*::GFP transcriptional reporter which did not produce detectable coelomocyte fluorescence [32]. This is consistent with the idea that FLP-18::Venus protein is secreted and taken up by coelomocytes, which are endocytic scavenger cells that internalize material from the pseudo-coelomic fluid [55]. We measured changes in fluorescent intensity in the AVA and RIM neurons after repeated activation of the escape response (Fig 4A and 4C). Mechanical tap to the plate caused a progressive reduction in FLP-18::Venus fluorescent intensity in the AVA and RIM neurons compared to animals that did not receive the tap stimuli (Fig 4D and 4E), suggesting that activation of the escape response elicits the release of FLP-18 from the AVA and RIM. Tap treatment did not change the fluorescent intensity in coelomocytes (Fig 4F). This could be due to protein break down in the coelomocytes and/or the high basal levels of coelomocyte fluorescence due to tonic FLP-18 release from other neurons.

### FLP-18's modulation of the escape response depends on *npr-5*

How does FLP-18 signaling modulate locomotion and the escape response? FLP-18 peptides activate the GPCRs: NPR-1, NPR-4 and NPR-5 *in vitro* [32,36,44]. We analyzed *npr-1*, *npr-4*, and *npr-5* loss-of-function mutants for defects in the escape response. *npr-1*, *npr-4* and *npr-5* mutants displayed largely normal locomotion and initiated an escape response upon anterior touch. While loss of *npr-1* or *npr-4* had no impact on the frequency of omega turns, *npr-5* mutants executed fewer omega turns compared to the wild type (Fig 5A). The decrease in omega turning of *npr-5* mutants was comparable to *flp-18* mutants. Furthermore, like *flp-18* mutants, *npr-5* mutants also display a higher proportion of open omega turns (Fig 5B). This suggests that FLP-18 peptides released during the escape response enhance head and body bending to promote turning through activation of the GPCR NPR-5.

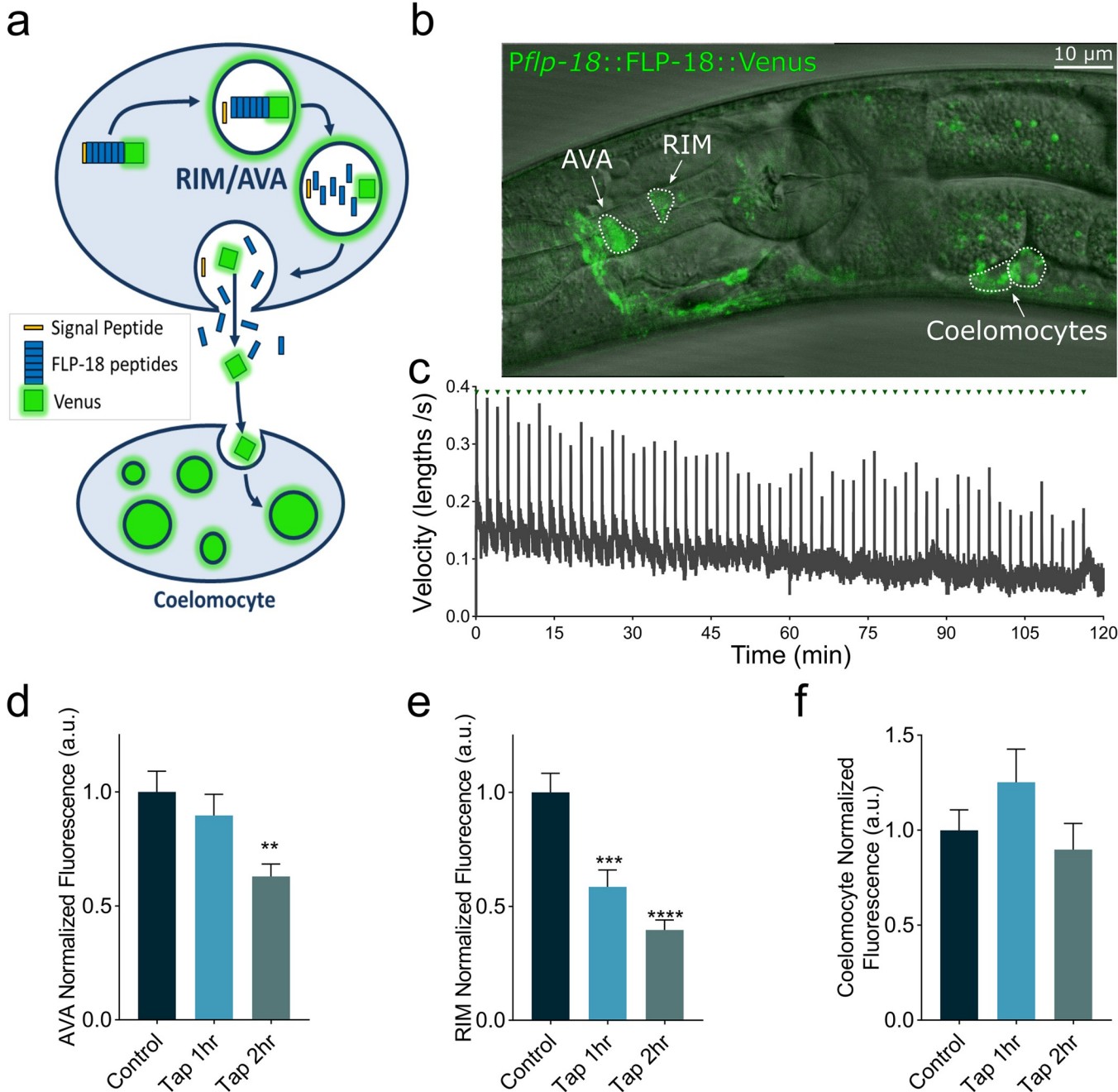

**Fig 4. The escape response stimulates FLP-18 release from the AVA and RIM.** Animals expressing a FLP-18::Venus fusion protein were subjected to repeated mechanical plate taps. The decrease in fluorescent intensity of the RIM and AVA cell bodies and the anterior-most coelomocyte were quantified as a readout of FLP-18 release. (A) Schematic illustrating the processing and exocytosis of the FLP-18::Venus fusion protein. (B) Representative image showing an animal expressing FLP-18::Venus, DIC and GFP overlay. Scale bar represents 10 μm. (C) Average velocity of a population of animals as they are subjected to mechanical plate taps. Green arrows indicate the delivery of a tap every 2 min. The spikes in velocity coincide with the initiation of the escape response (D-E) Quantification of fluorescence (arbitrary units–a.u.) in the AVA (D), RIM (E), and coelomocytes (F), at 0-, 1- and 2-hour time points. Graphs represent mean ± SEM, significance was calculated using ANOVA with Dunnett's multiple comparison test (P<0.005 = **, P<0.0005 = ***, P<0.0001 = ****). Sample size: (C) n = 3 replicates, >30 animals per replicate. (D-F) Control (n = 32), Tap 1 hr. (n = 26), Tap 2 hr. (n = 35).

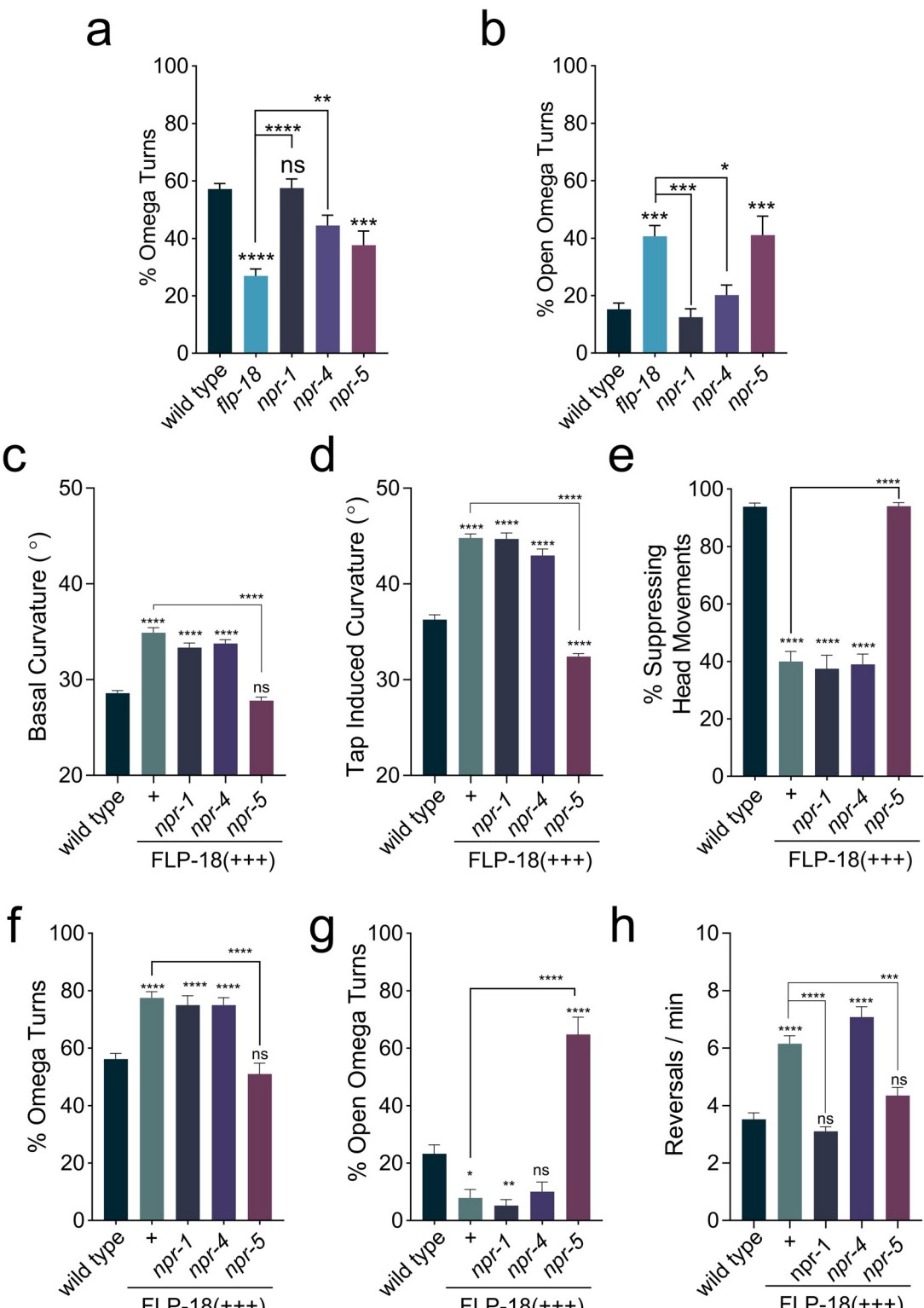

**Fig 5. Loss of *npr-5* impedes omega turning and suppresses FLP-18 overexpression phenotypes.** Overexpression of FLP-18 causes uncoordinated locomotion, frequent reversals, and excessive bending, which is suppressed in an *npr-5* mutant background. (A-B) Quantification of omega turning behavior. (A) Percentage of animals that execute omega turns in response to gentle anterior touch. (B) Fraction of open omega turns out of total omega turns. (C and D) Mean body curvature averaged over 5 seconds prior to a tap stimulus (C) or immediately following a strong tap stimulus (D). (E) Percentage of animals

suppressing head movements in response to gentle anterior touch with an eyelash. (F and G) Omega turning behavior in FLP-18 overexpressors. (F) Percentage of animals initiating omega turns after gentle anterior touch. (G) Proportion of open omega turns out of total omega turns. (H) Mean number of spontaneous reversals per minute per worm averaged over 3 minutes. Graphs represent mean ± SEM, significance was calculated using ANOVA with Šidák's multiple comparison correction (P> 0.05 = ns, P<0.05 = *, P<0.005 = **, P<0.0005 = ***, P<0.0001 = ****). Sample sizes (n = # of experiments, 20 animals per experiment): Omega turn quantification (A and B) wild type (n = 21), *flp-18* (n = 21), *npr-1* (n = 12), *npr-4* (n = 14), *npr-5* (n = 21). Curvature and reversal measurements (C, D, and H) wild type (n = 46), FLP-18(+++) (n = 33), *npr-1;* FLP-18(+++) (n = 23), *npr-4;* FLP-18(+++) (n = 27), *npr-5;* FLP-18(+++) (n = 23). Suppression of head movements and omega turn quantification (E-G) wild type (n = 22), FLP-18(+++) (n = 10), *npr-1*; FLP-18(+++) (n = 12), *npr-4*; FLP-18(+++) (n = 10), *npr-5;* FLP-18(+++) (n = 10).

Next, we examined whether neuropeptide receptor mutants could suppress the hyper bending phenotype of the FLP-18 overexpression strain. Mutations in either *npr-1*, *npr-4* or *npr-5* alone had no effect on body curvature during basal locomotion (S4A Fig). In the FLP-18(+++) background, deletion of *npr-5*, but not *npr-1* or *npr-4*, restored body curvature to wild-type levels (Fig 5C). Loss of *npr-5* also suppressed the hyper bending during the omega turn caused by FLP-18 overexpression (Figs 5D and S4B). *npr-5*; FLP-18(+++) animals make omega turns similar to wild type (Fig 5F). However, the proportion of open omega turns made by *npr-5;* FLP-18(+++) animals is significantly increased and similar to *npr-5* single mutants (Fig 5G).

*npr-1;* FLP-18(+++) and *npr-4;* FLP-18(+++) were indistinguishable from FLP-18(+++) in their failure to suppress head oscillations in response to touch. In contrast, *npr-5;* FLP-18(+++) animals suppressed head movements in response to anterior touch like wild-type animals (Fig 5E). Loss of both *npr-1* and *npr-5* reduced the high spontaneous reversal frequency caused by FLP-18 overexpression (Fig 5H). *npr-1* single mutants displayed significantly fewer spontaneous reversals compared to wild-type animals (S4C Fig), likely due to the fact that *npr-1* mutants avoid ambient oxygen concentrations [56]. Under low oxygen conditions loss of *npr-1* failed to suppress the increased reversal frequency caused by FLP-18 overexpression (S4D Fig). This suggests that the effect of *npr-1* on reversal frequency reflects its role in oxygen sensation independent of *npr-5*. We further analyzed *npr-5* mutants for locomotion defects during the escape response. Tap-stimulated peak reversal velocity was slightly reduced in *npr-5* mutants. In addition, there was a marked decrease in forward run speed (Fig 6A–6D), closely resembling the defect observed in *flp-18* mutants. Taken together these data indicate that FLP-18 modulates locomotory arousal in the escape response primarily through the activation of the NPR-5 receptor.

## NPR-5 acts in head-, neck and body wall muscle to drive body bending

To analyze where NPR-5 acts we generated transgenic animals that express a GFP tagged NPR-5 translational fusion [P*npr-5*::NPR-5::GFP]. NPR-5::GFP expression was detected in head, neck, and body-wall muscle (Fig 6E), and in a few neuronal cell bodies in the head (Fig 6E, lower right panel), consistent with previously described transcriptional reporters for *npr-5* [36]. NPR-5::GFP expression was particularly strong in head- and neck-muscles. We found that NPR-5::GFP localizes to dense bodies within muscle cells and is highly enriched in muscle arms and the muscle plate around the nerve ring (Fig 6E). The neck muscle arms around the nerve ring receive direct synaptic inputs from the FLP-18 expressing RIM neurons [20]. Overexpression of NPR-5 under control of the endogenous promoter (P*npr-5*::NPR-5) did not cause elevated curvature during basal locomotion (Fig 6F). However, in response to a mechanical stimulus, P*npr-5*::NPR-5 animals displayed increased body curvature comparable to FLP-18(+++) animals (Fig 6G). This suggests that the effect of NPR-5 overexpression driven by the endogenous promoter is uncovered when FLP-18 is released during the escape response. Cell-specific overexpression of NPR-5 in all body wall muscles (including head- and neck-muscles;

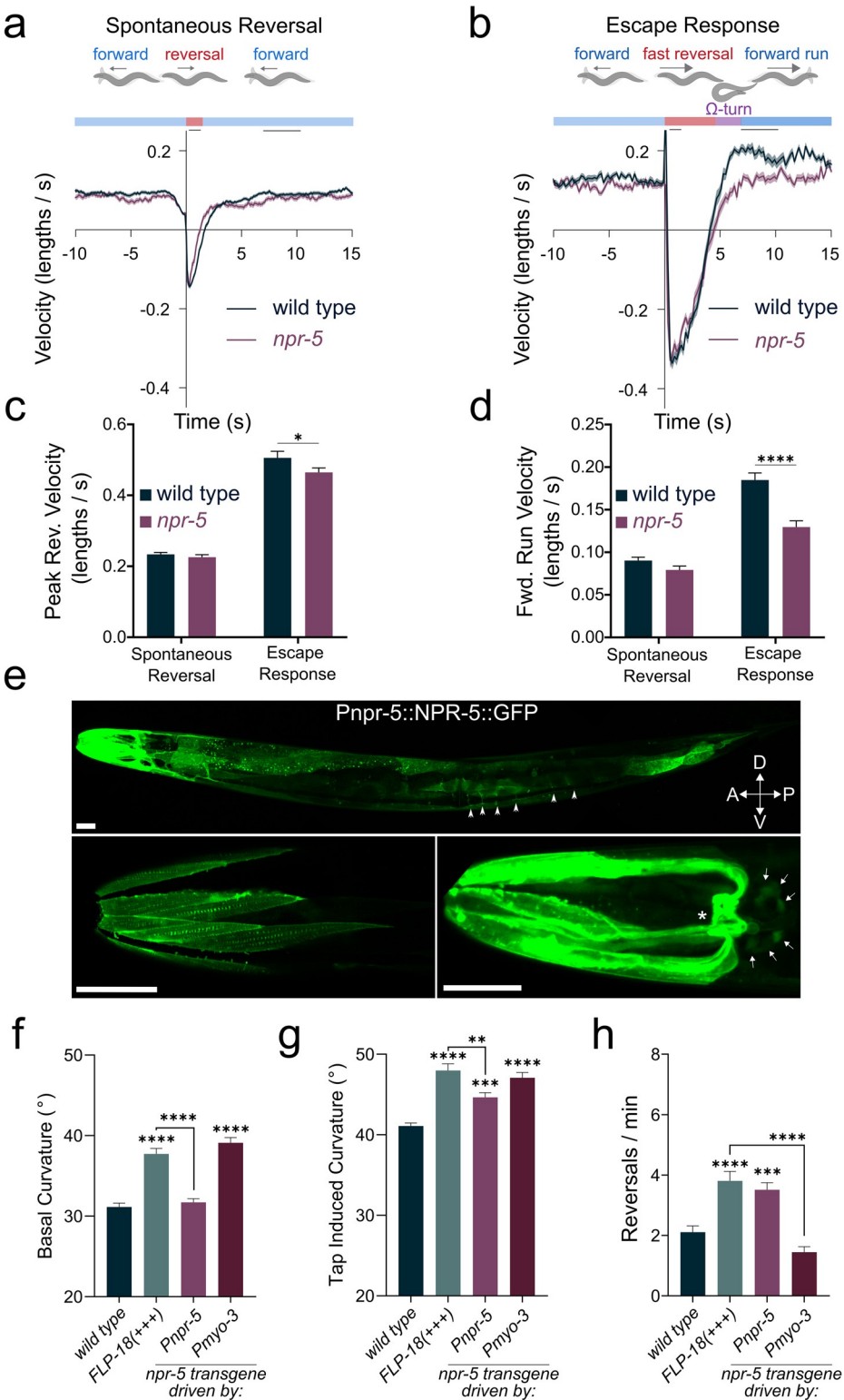

**Fig 6. *npr-5* mutants have defects in forward run speed, body bending and reversal frequency.** (A-B) Schematic illustrating behavioral sequence (top) and velocity traces (bottom) from wild type and *npr-5* mutants aligned to reversal events, backward locomotion is negative. Dark lines represent mean velocity and shaded region represents standard error. Reversals were spontaneous (A) or induced by a tap stimulus (B). (C) Peak reversal velocity during the first second following the tap stimulus. (D) Mean locomotion velocity during the forward run phase (t = 7-10s). (E)

Confocal images of animals expressing an NPR-5::GFP translational fusion protein: *zfEx852* [Pnpr-5::NPR-5::GFP]. Upper panel: Z-projection shows strongest expression in head- and neck-muscles and muscle arms. NPR-5::GFP localizes to body-wall and vulval muscle. Body wall muscle arms are visible (arrow heads). Lower left panel: confocal slice showing NPR-5::GFP localization to dense bodies in head muscle. Lower right panel: Z-projection showing high expression in head- and neck- muscle arms and in the muscle plate in the nerve ring (asterisk). Neuronal cell bodies that express NPR-5::GFP are indicated by arrows. Scale bars represent 25 μm. (F-H) Behavioral analysis of NPR-5 overexpression, either from its endogenous promoter (*Pnpr-5*) or from a muscle specific promoter (*Pmyo-3*). Mean body curvature averaged over 5 seconds prior to (F), or immediately following (G), a tap stimulus. (H) Quantification of spontaneous reversal frequency. Graphs represent mean ± SEM, significance was calculated using ANOVA with Šidák's multiple comparison correction (P>0.05 = ns, P<0.005 = **, P<0.0005 = ***, P<0.0001 = ****). Sample sizes: (A-C) (n = # of animals), wild type (n = 512), *npr-5* (n = 258). (F-H) (n = # of experiments, 20 animals per experiment). wild type (n = 20), FLP-18(+++) (n = 13), *Pnpr-5*::NPR-5 (n = 12), *Pmyo-3*::NPR-5 (n = 14).

*Pmyo-3*::NPR-5), on the other hand, did increase body curvature during both basal and tap-induced locomotion, similar to FLP-18 overexpressors (Fig 6F and 6G). The discrepancy in basal curvature between the *Pnpr-5*::NPR-5 and *Pmyo-3*::NPR-5 transgenic animals may be due to differences in NPR-5 expression pattern and/or level. *Pmyo-3* is expressed strongly in all body wall muscle, while expression of the endogenous *Pnpr-5* promoter is restricted to a subset of neurons and is high in head- and neck-muscles. Overexpression of NPR-5 under control of its endogenous promotor increased spontaneous reversal frequency, similar to FLP-18 overexpressing animals. Muscle specific expression of NPR-5 did not increase reversals (Fig 6H). Consistent with previous observations, this may suggest that neuronal expression of NPR-5 modulates reversal behavior [42]. Our results indicate that FLP-18 acts through NPR-5 in muscle to enhance body bending.

## FLP-18 increases calcium levels in body wall muscle

How does FLP-18 stimulate body bending? Muscle contractions are triggered by the rapid entry of calcium into the muscle cytosol [57,58]. To determine if FLP-18 signaling affects calcium levels in body wall muscle we generated a strain expressing the fluorescent calcium indicator GCaMP6s in body wall muscles [P*myo-3*::GcaMP6] (Fig 7A). *C. elegans* locomotion is driven by alternating waves of contraction and relaxation in dorsal and ventral muscles. During locomotion the highest GCaMP fluorescence was observed when muscles were contracted and faded upon relaxation, consistent with calcium influx driving muscle contraction (Fig 7A). *flp-18* mutants showed a reduction in GCaMP fluorescence in the muscle (Fig 7B). In sharp contrast, overexpression of FLP-18 resulted in a striking increase of GCaMP fluorescence in body-wall muscles (Fig 7A and 7B). This indicates that FLP-18 increases calcium transients in muscles that facilitate body bending.

Like *flp-18* mutants, *npr-5* mutants displayed a similar decrease in GCaMP fluorescence in body wall muscles. Furthermore, an *npr-5* mutation completely suppressed the increase in GCaMP fluorescence of FLP-18 overexpression (Fig 7B). *In vitro* experiments indicate that NPR-5 signals primarily through the $G\alpha_q$ pathway [44]. Activation of $G\alpha_q$ coupled receptors can cause the release of calcium from intracellular stores [59,60]. *egl-30* encodes the main *C. elegans* $G\alpha_q$ ortholog [61]. To determine whether *in vivo* NPR-5 signaling is $G\alpha_q$ dependent we examined the effect of mutations in *egl-30* on muscle calcium levels. Loss of *egl-30* substantially decreased GCaMP fluorescence in muscle. GCaMP fluorescence was also strongly reduced in the muscles of *egl-30;* FLP-18(+++) animals and indistinguishable from *egl-30* single mutants (Fig 7B). Mutations in FLP-18 signaling genes did not affect intensity of a co-expressed $Ca^{2+}$ insensitive fluorophore (S5A and S5B Fig) indicating that the differences in GCaMP fluorescence are due to altered intracellular $Ca^{2+}$ levels. Our results support the idea

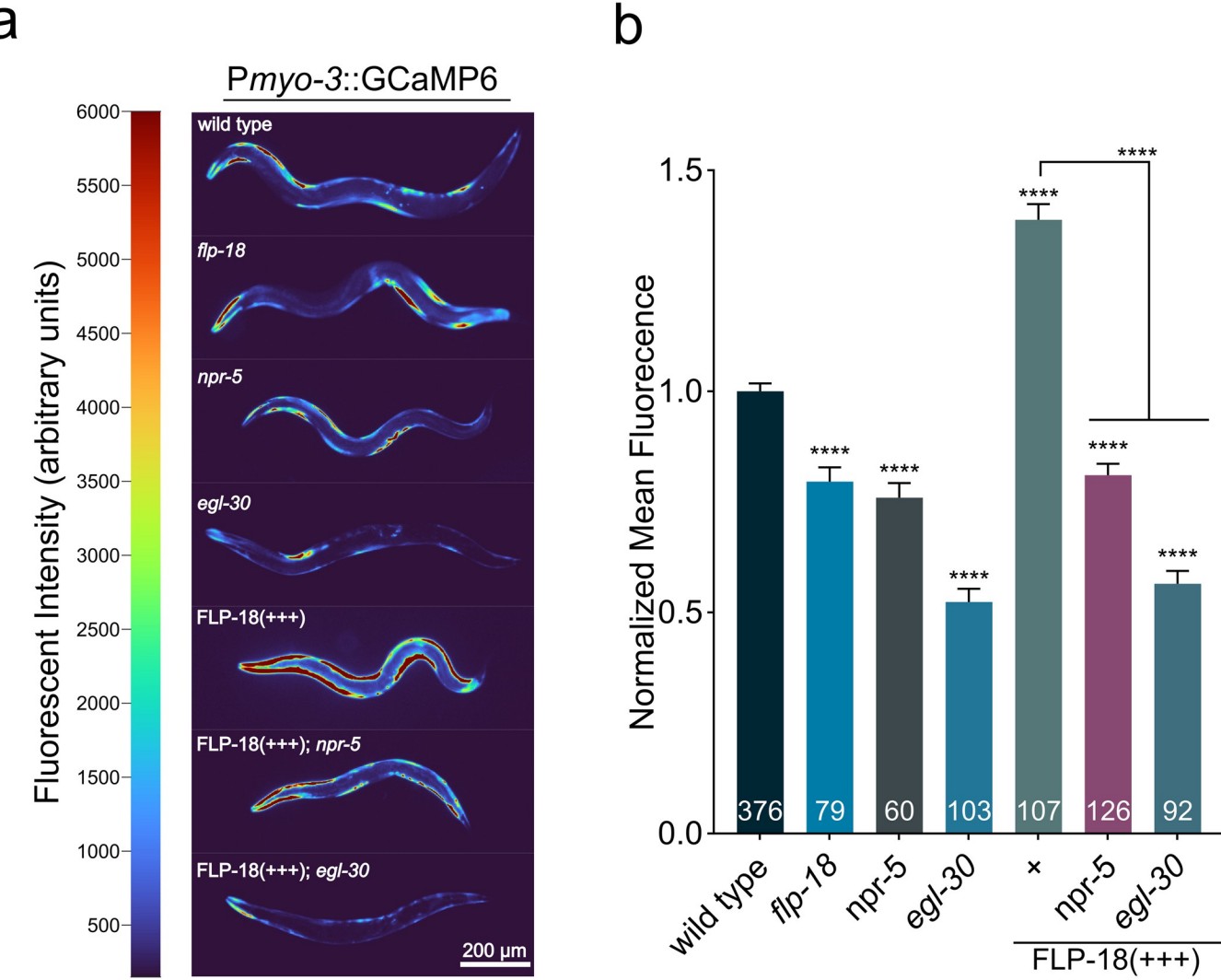

**Fig 7. FLP-18 increases muscle calcium levels and requires NPR-5 and EGL-30/Gα$_q$.** (A) Loss of *flp-18* or *npr-5* reduces calcium transients in body wall muscle. FLP-18 overexpression causes a large increase in muscle GCaMP6 fluorescence, which is suppressed by loss of *npr-5* or *egl-30*. Representative images of animals expressing a P*myo-3*::GCaMP6 transgene in different genetic backgrounds. Grayscale images have been recolored for visibility. Pixel intensity values (arbitrary units) and corresponding colormap are depicted in the color bar to the left of images. Scale bar represents 200 μm. (B) Quantification of mean GCaMP6 fluorescence in body wall muscle in different genetic backgrounds. Graphs represent mean ± SEM, significance was calculated using ANOVA with Šidák's multiple comparison correction (P<0.0001 = ****). Sample size for each genotype is indicated at the base of each bar.

that FLP-18 activates NPR-5 and EGL-30/Gα$_q$ signaling cascades, resulting in increased calcium levels in the muscle (Fig 8B).

## Discussion

### FLP-18 enhances motor output through the activation of NPR-5

In this study we identify FMRFamide-like peptides encoded by the *flp-18* gene as key modulators of the distinct phases in the *C. elegans* escape response. We showed that FLP-18 neuropeptides are released from the RIM and AVA neurons during an escape response elicited by a mechanical stimulus. FLP-18 promotes head and body bending during the omega turn and increases forward run velocity following the turn. We find that the effects of FLP-18 on

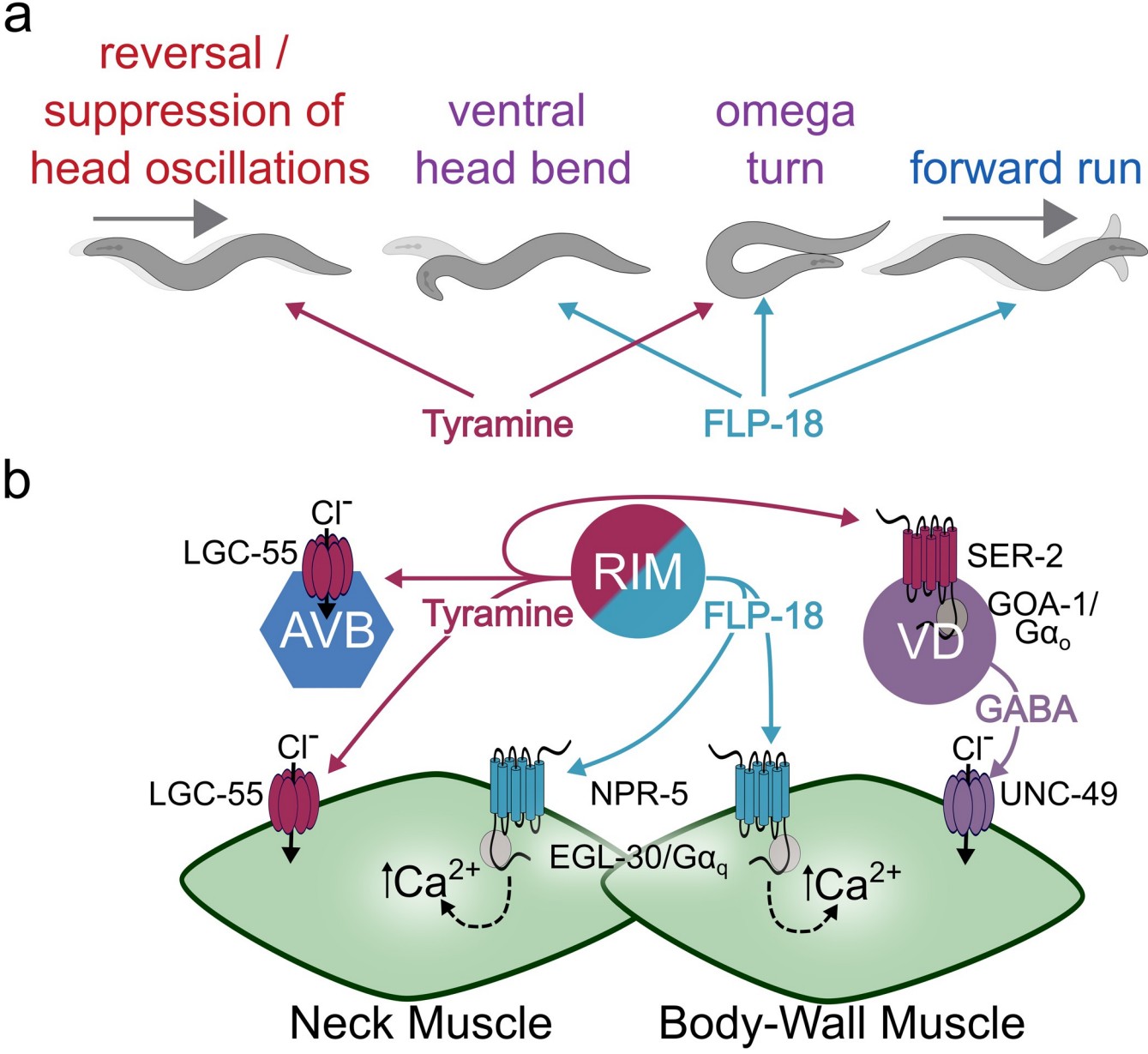

**Fig 8. Model: FLP-18 / tyramine co-transmission in the choreography of escape behavior.** Schematic depicting the effects of tyramine and FLP-18 on behavior (A) and the underlying molecular pathway (B). Touch leads to the activation of the AVA (not pictured) and RIM neurons resulting in the release of FLP-18 and tyramine (RIM only) during the escape response. Tyramine activates LGC-55 chloride channels which hyperpolarize AVB neurons and neck muscle. Hyperpolarization of the AVB suppresses forward locomotion, indirectly increasing reversal length, while hyperpolarization of neck muscle leads to the suppression of head oscillations (SHO) during the reversal. Tyramine also activates the $G\alpha_o$ coupled SER-2 receptor in GABAergic VD motor neurons enhancing ventral bending during the omega turn. FLP-18 activates NPR-5 in neck and body wall muscle resulting in an EGL-30/$G\alpha_q$ mediated $Ca^{2+}$ influx. FLP-18 antagonizes the effect of tyramine in neck muscles, facilitating the re-initiation of head movements following the omega turn. FLP-18 and tyramine act synergistically on divergent targets to orchestrate the omega turn. FLP-18 increases forward run velocity in an NPR-5 dependent manner. See text for details.

locomotion are largely mediated through activation of the $G\alpha_q$ coupled NPR-5 receptor. NPR-5 is highly expressed in head and body wall muscles which receive direct synaptic inputs from the RIM neurons. Studies in both vertebrates and *C. elegans* have shown that EGL-30/$G\alpha_q$ activates phospholipase Cβ (PLCβ) [62,63]. EGL-8/PLCβ increases IP$_3$ levels which can lead to the

release of $Ca^{2+}$ from internal stores [64,65]. We find that FLP-18 activation of NPR-5 increases calcium levels in muscles *in vivo* in an EGL-30/$G\alpha_q$ dependent manner. *In vitro* studies that show that NPR-5 activation by FLP-18 peptides resulted in rises in intracellular $Ca^{2+}$ levels [44]. Our data support a model in which FLP-18 release during the flight response stimulates $Ca^{2+}$ release in head and body wall muscle through the NPR-5—EGL-30/$G\alpha_q$ signaling pathway (Fig 8B). Increases in calcium levels would stimulate muscle tension, required for head and body bending the during the omega turn.

Previous studies have indicated that FLP-18 acts in part through NPR-5 in body wall muscles to decrease the convulsion phenotype of nicotinic acetylcholine receptor *acr-2(gf)* mutants [39]. Our data raise the possibility that a FLP-18 dependent increase in intracellular $Ca^{2+}$ concentrations may reduce the $Ca^{2+}$ gradient across the muscle membrane and thus provides a homeostatic response in muscles to the elevated neuronal activity of cholinergic motor-neurons in *acr-2(gf)* mutants. Recent reports have shown FLP-18 acts through NPR-5 in sensory neurons to control reversal frequency [42]. We found that NPR-5 overexpression using the endogenous promoter (neurons and muscle) increases reversal frequency, while muscle-specific overexpression had no effect. This observation is consistent with a neuronal contribution of NPR-5 in the modulation of reversals. We hypothesize that FLP-18 and NPR-5 mediated increases in muscle $Ca^{2+}$ levels and excitability contribute to the increase in forward run velocity following the turn. However, we cannot exclude a role for neuronal FLP-18/NPR-5 signaling in the regulation of locomotion rate.

Together, our results show that FLP-18 signaling functions to enhance motor output during the escape response and locomotory arousal through activation of NPR-5 in muscle. FLP-18 thus may contribute a locomotory arousal state that enhances vigilance and the ability to escape from predation. Other studies have shown that FLP-18 activation of NPR-1, dampens the activity of sensory neurons to promote the sleep like quiescent period associated with larval molts [37,38,66,67]. Interestingly, activation of the escape response also induces the release of a distinct RFamide peptide, FLP-20, from mechanosensory neurons. FLP-20 release enhances arousal and motor output across multiple sensory modalities through activation of NPYR like receptor FRPR-3 in the neurosecretory RID neurons [46]. These findings highlight the multilevel nature of RFamide neuropeptides in the modulation of arousal in *C. elegans*.

## Co-transmission of FLP-18 and tyramine orchestrate the flight response

Many neurons contain both small molecule transmitters and neuropeptides. In addition to FLP-18, the RIM neurons also contain additional neuropeptides and the neuromodulator/neurotransmitter tyramine and glutamate [26,68–70]. We have previously shown that tyramine release from the RIM during the escape response coordinates independent motor programs of the escape response [26,28,29]. Our findings indicate that activation of the escape response results in the co-transmission of tyramine and FLP-18 from the RIM. The RIM neurons contain dense core vesicles [71] which can store both neuropeptides and monoamines [72,73]. The RIM also contains regular synaptic vesicles (possibly for the release of glutamate); thus, it is currently unclear whether tyramine and FLP-18 are co-released from the same vesicle. The well-defined escape circuit and the role of the RIM in this response provides the opportunity to dissect of how tyramine and FLP-18 co-transmission choreograph independent motor outputs in the execution of the escape response. Co-transmission of FLP-18 and tyramine from the RIM are convergent in some aspects of the escape response but divergent in others. For example, FLP-18 and tyramine both act to facilitate deep bending during the omega turn but do so through distinct mechanisms in different tissues. Tyramine inhibits the GABAergic VD motor-neurons through activation of the $G\alpha_o$ coupled receptor SER-2 [29].

The inhibition of GABA release onto ventral body wall muscle facilitates the deep ventral contraction during the omega turn. FLP-18 on the other hand activates NPR-5 in the neck- and body wall-muscles resulting in increased calcium levels and enhanced head and body bending. Thus, co-transmission of FLP-18 and tyramine exert excitatory and inhibitory effects in different cells resulting in functional synergism that enhances omega bending (Fig 8A and 8B). In contrast to the complementary function of tyramine and FLP-18 on body bending, we observe convergent but antagonistic effects of these neuromodulators on exploratory head oscillations. During the reversal phase of the escape response, wild-type *C. elegans* suppress head oscillations to increase its chances to escape encounters with hyphal nooses of predatory fungi [27]. Tyramine release from the RIM mediates the suppression of head oscillations through the activation of a tyramine-gated chloride channel, LGC-55, in neck muscles and cholinergic motor neurons that synapse onto neck muscles [28]. Animals that overexpress FLP-18 fail to suppress head oscillations during reversals, but this defect is rescued by loss of *npr-5*. The simplest explanation is that in FLP-18 overexpressing animals, the tyramine induced hyperpolarization of neck muscles is no longer sufficient to overcome the excess of FLP-18 induced depolarization. In the behavioral sequence of the escape response, head oscillations are reinitiated during the forward run that follows the omega turn. The tyraminergic activation of a fast-acting ionotropic LGC-55 receptor and FLP-18 activation of the slow-acting metabotropic NPR-5 receptor may contribute to coordinate the timing of suppression and re-initiation of head oscillations.

The use of genetically tractable model organisms with simple nervous systems like *C. elegans* allows systematic interrogation of the mechanisms underlying co-transmitter modulation of behavior. Our work has shown examples of convergent/antagonistic as well as divergent/complementary actions of co-transmitters released from a single neuron pair. Using a similar genetic and behavioral approach, other functional properties of co-transmission have been elucidated in *C. elegans*. Co-transmission of serotonin and the neuropeptide NLP-3 coordinate the activity of the *C. elegans* egg-laying circuit [74]. Foraging behavior and olfactory adaptation are shaped by convergent / antagonistic co-transmission of glutamate and NLP-1 from the AWC sensory neurons to dampen olfactory responses [75].

The complexity of mammalian system makes it more difficult to dissect the in vivo consequences of neuropeptide and co-transmission. Nevertheless, there are intriguing similarities in mammals where neuropeptide Y (NPY) is co-released with noradrenaline (NA) in response to stress [6,7,15,16,76,77]. In mammals, co-transmission of NA and NPY in the mammalian sympathetic nervous system has both complementary and antagonistic actions. For example, both NA and NPY produce a pressor response through contraction in vascular smooth muscle [10,12,16], while NPY and NA have opposing inotropic and chronotropic effects on cardiac muscle [11,13,14,78]. The similarities between vertebrate NA/NPY and *C. elegans* tyramine/FLP-18 signaling indicate that co-transmission monoamines and neuropeptides is a central feature of the signaling pathways that underlie arousal in response to stress. In the presence of danger, heightened arousal can prepare an animal to respond more effectively to a threat and increase the chance of survival. The stereotyped *C. elegans* escape response allows us to elucidate how co-transmission from a single neuron can shape circuit activity on different time scales to orchestrate arousal and physiology in response to stress.

## Materials and methods

### Molecular biology and transgenics

Standard molecular biology methods were used. A list of strains used is this manuscript is included as Table 1. Most transgenic strains were made through microinjection of plasmid

**Table 1. Strain List.** A list of *C. elegans* strains used in this manuscript.

| | |
|---|---|
| N2 | Wild Type |
| CX4148 | *npr-1*(ky13) |
| QW940 | *npr-4*(tm1782) *4x backcrossed from FX1782* |
| QW953 | *npr-5*(ok1583) *4x backcrossed from RB1393* |
| QW1223 | *flp-18(tm2179)*; zfex528[Pcex-1::FLP-18::SL2::mCherry] (injected at 100ng/µl) |
| QW1379 | *flp-18*(gk3063) *4x backcrossed from VC2016* |
| QW1526 | *lin-15(n765ts)*, zfex713[Pnpr-5::NPR-5 unc-122::RFP +PL15EK] (injected at 200ng/µl) |
| QW1646 | *lin-15(n765ts)*; zfex813[Pmyo-3::NLSwCherry::SL2::GCaMP6 +PL15EK] (injected at 100ng/µl) |
| QW1655 | [FLP-18(+++)] *lin-15(n765ts)*; zfis149[Pflp-18(3kb)::mCherry::SL2::FLP-18 +PL15EK] X. (injected at 200ng/µl) 4x backcrossed. |
| QW1675 | *npr-4(tm1782)*; zfis149[Pflp-18(3kb)::mCherry::SL2::FLP-18 +PL15EK] X. |
| QW1677 | zfex813[Pmyo-3::NLSwCherry::SL2::GCaMP6]; zfis149[Pflp-18(3kb)::mCherry::SL2::FLP-18 +PL15EK] X. |
| QW1680 | [FLP-18(+)] *flp-18(gk3063)*; zfex821[pflp-18::FLP-18, +unc-122::GFP] (injected at 20ng/µl) |
| QW1714 | *lin-15(n765ts)*, zfex837[pflp-18::FLP-18::Venus +PL15EK] (injected at 200ng/µl) |
| QW1754 | *lin-15(n765ts)*; zfex845[Pmyo-3::NPR-5::SL2::mCherry +PL15EK] (injected at 200ng/µl) |
| QW1770 | *lin-15(n765ts)*; zfex852[Pnpr-5::NPR-5::GFP +PL15EK] (injected at 200ng/µl) |
| QW1784 | *npr-5(ok1583)*; zfex813[Pmyo-3::NLSwCherry::SL2::GCaMP6]; zfis149[Pflp-18(3kb)::mCherry::SL2::FLP-18 +PL15EK] |
| QW1802 | *flp-18(gk3063)*; zfex813[Pmyo-3::NLSwCherry::SL2::GCaMP6] |
| QW1818 | *npr-5(ok1583)*; zfis149[Pflp-18(3kb)::mCherry::SL2::FLP-18 +PL15EK] |
| QW1819 | *npr-5(ok1583)*; zfex813[pmyo-3::NLSwcherry::SL2::GCamp6] |
| QW1827 | *egl-30(ad805)*; zfex813[Pmyo-3::NLSwCherry::SL2::GCaMP6]; zfis149[Pflp-18(3kb)::mCherry::SL2::FLP-18 +PL15EK] |
| QW1855 | *egl-30(ad805)*; zfex813[Pmyo-3::NLSwCherry::SL2::GCaMP6] |
| QW2169 | *npr-1(ky13)*; zfis149[Pflp-18(3kb)::mCherry::SL2::FLP-18 +PL15EK] |

DNA into *lin-15(n765ts)*. A *lin-15* rescuing plasmid (PL15EK) was used as a co-injection marker (80 ng/µl) and transgenics were identified based on rescue of the *lin-15* multivulva phenotype. FLP-18 transgenic strains were created by amplifying a 5.9kb sequence of genomic DNA beginning 3.3kb upstream of the open reading frame and containing the *flp-18* coding region as well as the 3' UTR. The sequence was amplified using the forward primer 5'-GTG ATGTAGGTAGCGCGGAAC-3' and 5'-TTCATGATCCCAGATTCAATTAATC-3' reverse primer. This amplicon was ligated into a plasmid containing an SL2 linked mCherry transcriptional reporter to visualize expression. This plasmid was injected along with pL15EK and transgenics were identified based on *lin-15* rescue. The extrachromosomal array was stably integrated into the genome using X-ray irradiation and were backcrossed to wild type N2 animals 4 times. NPR-5 transgenics were created using Gibson Assembly (Gibson Assembly Master Mix (E2611), New England BioLabs inc., [79]) to insert the *npr-5* genomic locus into a pSP72 plasmid backbone. The insert consisted of a 7.7kb sequence of genomic DNA beginning 3kb upstream of the *npr-5* open reading frame and including the *npr-5* coding sequence and 3' UTR. The sequence was amplified using 5'-cagatctgatatcatcgatGGCCCCGTGATTCTTGTC A-3' as a forward primer and 5'-atccccgggtaccgagctcgaatTCTTGCTCGGTGTGCATTTTC-3' as a reverse primer. The lowercase letters indicate the homology arms that anneal to pSP72 sequence, and the uppercase letters anneal to the *npr-5* genomic locus for amplification. pSP72 was linearized for insertion using EcoRI. The resulting plasmid was injected along with pL15EK and transgenics were identified based on *lin-15* rescue.

## Behavioral experiments

All behavioral experiments were carried out using young adult animals which had been staged as L4s the previous day. Behavioral experiments were performed on medium (60mm) thin-lawn NGM plates where a single drop of OP50 (~30μl) was spread across the plate and allowed to grow at room temperature overnight. Thin lawn plates older than 24hrs were discarded. Prior to behavioral experiments, animals were transferred from standard plates to thin lawn plates and allowed to acclimate for at least 5 minutes. All experiments were conducted with 20 worms to a plate unless otherwise stated and were run in parallel with wild type controls across at least 3 days.

## Worm tracking

Quantification of locomotion was performed using one of two different worm tracking systems. For quantification of foraging speed (Fig 2C), the Worm Tracker 2.0 system was used (Eviatar Yemini and Tadas Jucikas, https://www.mrc-lmb.cam.ac.uk/wormtracker, [80]) and "absolute foraging speed" was reported. A single animal on a thin lawn plate was tracked for 5 minutes at 30fps using a USB digital microscope (DinoLite Pro AM413T, Dino-Lite US) on a motorized stage (Zaber TNA08A50 linear actuator and Zaber TSB60-M motorized stage, Zaber Technologies Inc.).

Quantification of velocity, body curvature and reversal frequency were measured using the Multi-Worm Tracker (Rex Kerr, https://sourceforge.net/projects/mwt/). When tracking experiments included a tap stimulus, the stimulus was delivered by a tubular push solenoid (Saia-Burgess/Ledex #195205–127) controlled by the Multi-Worm Tracker software. The tap stimulus consisted of a train of 20 taps with an inter tap interval of 10 milliseconds. Tap stimuli were always delivered after 5 minutes of tracking had elapsed.

Experiments were analyzed using custom MATLAB (The MathWorks, Inc.) scripts to interface with the Multi-Worm Tracker feature extraction software Choreography. Analysis was limited to objects that had been tracked for a minimum of 20 seconds and had moved a minimum of 5 body lengths. All experiments were conducted on thin lawn plates with 20 animals on each plate.

## Body curvature

For comparisons of basal versus tap stimulated curvature animals were tracked using the Multi-Worm Tracker. The body curvature of the population was averaged over 5 seconds either just prior to the tap stimulus for basal curvature or just after the tap stimulus for stimulated curvature.

## Reversal frequency

20 animals were tracked while freely moving on a thin lawn plate with the Multi-Worm Tracker. The total number of reversals made by the population in a 3-minute window was extracted using the Multisensed plugin for Choreography (Rex Kerr) and divided by the average number of worms tracked during that time to get the number of reversals per worm in 3 minutes. For simplicity this value was divided by 3 and plotted as number of reversals per worm per minute.

## Touch assays

Experiments quantifying omega turning, suppression of head oscillations and ventral turn head angle were conducted by gently touching an animal just posterior to the pharynx with a

fine paint brush bristle taped to a glass pipette as previously described [47]. Animals were transferred to a thin lawn plate and allowed to acclimate for at least 5 minutes. 20 animals were assayed in each experiment and the population average was reported for each behavior. Animals were touched a single time and animals that failed to reverse were ignored. Animals that moved their head from side to side during a reversal were scored as defective in the suppression of head oscillations.

### Omega turns

Omega turns were defined as any turn which results in a reorientation greater than 90˚ from their initial forward bearing before the reversal. Turns that were greater than 90˚ where the animal failed to touch its body with its head were considered open omega turns. Percent omega turns for a given experiment was calculated as the number of open plus the number of closed turns divided by the total number of animals that reversed. The percentage of open omegas was calculated as the number of open omega turns divided by the total number of omega turns (open + closed) in that experiment.

### Ventral turn angle

To calculate the angle of the head during the initiation of a ventral turn animals were touched with an eyelash while video was being recorded through a dissecting microscope (SZ6TR1 Olympus) with a digital camera (AVT Pike F421-b, Allied Vision Technologies) using Fire-I image acquisition software (Unibrain inc.). Videos of animals that made omega turns were loaded into ImageJ / Fiji [81] and the first frame of forward movement following the touch initiated reversal was identified. To calculate the head deflection angle two lines were drawn, one running from the tip of the nose to the beginning of the intestine and the second running from the beginning of the intestine to a point on the midline 1 pharynx length further posterior. The angle between these two lines was measured using the angle tool in Fiji.

### Confocal imaging

Animals were immobilized with 60mM sodium azide and placed on a 2% agarose pad under a coverslip. Images were acquired using a Zeiss LSM700 confocal microscope (Carl Zeiss AG)

### Venus quantification

A population of animals expressing the *Pflp-18*::FLP-18::Venus transgene were subjected to a plate tap every 2 minutes for up to 2 hours. Individuals were removed at 0, 1 and 2 hours and immediately mounted and anesthetized as described above. Z-stacks were acquired with a confocal microscope (LSM 700, Carl Zeiss Microscopy GmbH) using a 40x objective (C-Apochromat 40x/1.20 W Korr, Carl Zeiss Microscopy GmbH). Images were loaded into ImageJ and the soma of the RIM and AVA were identified from their anatomical location. A circular region of interest (ROI) with a diameter of 13.5μm was placed around the soma or coelomocyte to measure the mean grey value of each neuron. An additional measurement was taken adjacent to the soma to quantify background fluorescence; this value was subtracted from the gray values of the RIM, AVA, and coelomocyte in that animal. The values obtained for each condition were normalized to the average intensity of the relevant neuron or coelomocyte in the control condition.

## Calcium imaging

Young adult animals expressing a *Pmyo-3*::GCaMP6::SL2::mCherry transgene were placed into a drop of M9 buffer on a 2% agarose pad atop a microscope slide and a coverslip was gently placed onto the pad. Animals were able to move freely under the coverslip and were imaged on an inverted fluorescent microscope (Axio Observer A.1, Carl Zeiss Microscopy GmbH) using a Hammamatsu ORCA-Flash4.0 camera within 10 minutes of mounting. Images were analyzed using ImageJ software by drawing a 1075μm x 630μm rectangular ROI which could be rotated or translated to fit the position of an animal and the mean gray value was measured. The same ROI was placed in a position containing no animals to measure the mean grey value of the background which was then subtracted from the grey value of the ROI containing the animal. Wild type animals expressing the *Pmyo-3*::GCaMP6::SL2::mCherry transgene were imaged in parallel with mutants and the average background-subtracted grey value of the wild type animals for that day was used to normalize the values obtained from imaging the mutants. mCherry imaging was performed in a similar manner using the same rectangular ROI and the raw pixel values were quantified.

## Supporting information

**S1 Fig. Escape behavior in *tdc-1;flp-18* double mutants.** (A and B) Behavioral schematic (top) and velocity traces (bottom) of animals executing the escape response. (A) In wild type animals, forward run velocity remains elevated for approximately 3 minutes after the escape response. Dotted line marks pre-stimulus baseline. (B) Comparison of *tdc-1;flp-18* double mutants to the velocity traces presented in Fig 1B. (A-B) Negative velocity indicates backward locomotion. Solid line in trace represents mean velocity, shaded area indicates standard error of the mean. (C-F) Quantification of escape behavior in *flp-18*, *tdc-1*, and *tdc-1;flp-18* mutant animals. See Fig 3 for details. Graphs represent mean ± SEM, significance was calculated using ANOVA with Šidák's multiple comparison correction (P<0.05 = *, P<0.005 = **, P<0.0005 = ***, P<0.0001 = ****). Sample size: (A) n = 14 independent trials with 20 worms recorded during each trial. (B) n = # of animals. Wild type, *flp-18*, and *tdc-1* values are reported in Fig 1. *tdc-1;flp-18* (n = 204). (C-E), n = # of experiments, 20 worms per experiment. Wild type (n = 12), *flp-18* (n = 11), *tdc-1* (n = 12), *tdc-1;flp-18* (n = 11). (F) n = # of animals. Wild type (n = 88), *flp-18* (n = 58), *tdc-1* (n = 60), *tdc-1;flp-18* (n = 60).
(TIF)

**S2 Fig. Expression pattern and behavioral analysis of *Pcex-1*::FLP-18::SL2::mCherry.** (A) Confocal z-projection of *zfEx528*[*Pcex-1*::FLP-18::SL2::mCherry showing detail of head neurons. Cell bodies are labeled, neurons were identified based on anatomical position. (B-D) Analysis of FLP-18 overexpression from endogenous (P*flp-18*) and cell specific (P*cex-1*) promotors. Spontaneous reversal frequency (B), mean body curvature averaged over 5 seconds prior to (C), or immediately following (D), a tap stimulus. Graphs represent mean ± SEM, significance was calculated using ANOVA with Šidák's multiple comparison correction (P<0.05 = *, P<0.0001 = ****). Sample size: (B-D) n = # of experiments, 20 animals per experiment. Wild type (n = 16), P*flp-18*::FLP-18(+++) (n = 4), Pcex-1::FLP-18(+++) (n = 6).
(TIF)

**S3 Fig. Animals overexpressing FLP-18 fail to suppress head oscillations during escape.** Minimum intensity projections from video recordings of wild type (A) and FLP-18 overexpressing animals (B) after anterior touch. "start" indicates location of head when the animal was touched and initiated a reversal. The omega (Ω) symbol indicates the point where an omega turn was initiated, and the animal switched from reversal to forward motion. "end"

indicates the position of the head at the end of the video. The head of the wild-type animal remains relaxed and follows the body during the reversal (A). The FLP-18(+++) overexpressing animal continues head oscillations during the reversal which are visible as the tip of the nose projecting out from the sides of the body and are indicated with red arrows (B). Scale bar = 100μm.
(TIF)

**S4 Fig. Characterization of reversal behavior and body curvature in FLP-18 receptor mutants.** (A and B) Quantification of mean body curvature averaged over 5 seconds prior to a tap stimulus (A) or immediately following a strong tap stimulus (B). (C and D) *npr-5* but not *npr-1* mutation suppresses high spontaneous reversal frequency in low oxygen conditions. Quantification of spontaneous reversal frequency in ambient oxygen (C) or in low oxygen conditions (D). Mean number of spontaneous reversals per minute per worm averaged over 3 minutes as in Fig 5H. Ambient oxygen experiments (C) were conducted with a standard lid-covered plate in room air. Low-oxygen experiments (D) were conducted on a plate that had nitrogen gas injected into it with a 5ml syringe prior to placing a lid. After lid placement, both conditions were allowed to acclimate for 5 minutes prior to recording. Graphs represent mean ± SEM, significance was calculated using ANOVA with Šidák's multiple comparison correction ($P<0.05 = *$, $P<0.005 = **$, $P<0.0001 = ****$). Sample sizes (# of experiments, 20 animals per experiment): (A-B) wild type (n = 22), *npr-1* (n = 14), *npr-4* (n = 12), *npr-5* (n = 13). (C) wild type (n = 26), *flp-18* (n = 28), *npr-1* (n = 14), *npr-4* (n = 23), *npr-5* (n = 17). (D) wild type (n = 12), FLP-18(+++) (n = 12), *npr-1; FLP-18(+++) (n = 15), *npr-5; FLP-18(+++) (n = 13).
(TIF)

**S5 Fig. Muscle::mCherry levels are unaffected by mutations in FLP-18 signaling.** Quantification of mCherry fluorescence in animals expressing the transgene *zfEx813*[P*myo-3*::NLSwCherry::SL2::GCaMP6] (muscle::GCaMP) in the genetic backgrounds analyzed in Fig 7B. Because the FLP-18 overexpression line (*zfIs149*[Pflp-18(3kb)::mCherry::SL2::FLP-18]) carries an mCherry transcriptional reporter, mutant lines are compared to their respective control background, muscle::GCaMP (A) or muscle::GCaMP; FLP-18(+++) (B). (C) Fluorescent micrograph showing mCherry, GCaMP and merged signals of a animals expressing *zfIs149* alone (bottom, FLP-18(+++)) or in combination with *zfEx813* (top, muscle::GCaMP; FLP-18(+++)). Scale bar = 200 μm. Graphs represent mean ± SEM, significance was calculated using ANOVA with Dunnett's multiple comparison correction ($P>0.05 = ns$). Sample sizes (# animals): (A) muscle::GCaMP (n = 46), *flp-18* (n = 30), *npr-5* (n = 23), *egl-30* (n = 15). (B) FLP-18(+++) (n = 39), *npr-5; FLP-18(+++) (n = 21), *egl-30; FLP-18(+++) (n = 30).
(TIF)

**S1 Data. Raw data used to generate all graphs in the manuscript.** The compressed folder contains 11 excel files each corresponding to a figure which is specified in the filename. Excel files are broken down into multiple sheets, each containing the data for an individual panel of a figure.
(ZIP)

**S1 Video. Locomotion of Pflp-18_FLP-18(+++): overexpression of FLP-18 driven by the endogenous promotor results in uncoordinated locomotion, deep body bends and increased reversals.**
(MP4)

**S2 Video. Locomotion of Pcex-1_FLP-18(+++): overexpression of FLP-18 from the *cex-1* drives expression in the RIM, AVA, and AVD and results in uncoordinated locomotion, deep body bends and increased reversals, similar to overexpression from the endogenous promoter.**
(MP4)

**S3 Video. Touch response in wild type.** The escape response to anterior touch in wild type animals consists of a reversal during which head movements are suppressed followed by an omega turn and resumption of head movements and forward locomotion in the opposite direction. An eyelash is used to gently touch the animal behind the head to trigger the escape response.
(MP4)

**S4 Video. Touch response in FLP-18(+++).** The escape response in animals overexpressing FLP-18 results in deeper body bends and a failure to suppress exploratory head movements while reversing. An eyelash is used to gently touch the animal behind the head to trigger the escape response.
(MP4)

## Acknowledgments

Some strains were provided by the CGC, which is funded by NIH Office of Research Infrastructure Programs (P40 OD010440). We thank Mario de Bono for strains and Rex Kerr for the multi-worm tracker software, Will Joyce for worm injections and Michael Francis for critically reading the manuscript, Daniel Witvliet and Mei Zhen for sharing RIM electron microscopy images.

## Author Contributions

**Conceptualization:** Mark J. Alkema.

**Funding acquisition:** Mark J. Alkema.

**Investigation:** Jeremy T. Florman.

**Writing – original draft:** Jeremy T. Florman.

**Writing – review & editing:** Mark J. Alkema.

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
