## [Decision Letter · Decision Letter 0]

16 Nov 2021

Dear Dr %Alkema%,

Thank you very much for submitting your Research Article entitled 'Co-transmission of neuropeptides and monoamines choreograph the C. elegans escape response.' to PLOS Genetics.

The manuscript was fully evaluated at the editorial level and by independent peer reviewers. The reviewers appreciated the attention to an important topic but identified some concerns that we ask you address in a revised manuscript

We therefore ask you to modify the manuscript according to the review recommendations. Your revisions should address the specific points made by each reviewer.  We believe that most of the concerns of each of the three reviewers can addressed through textual revisions.  However, as you can see in their detailed comments, the reviewers have asked for the inclusion of several additional experiments that are necessary for clear interpretation of the results. 

We note that in several specific instances the reviewers have suggested revisions or additions that will strengthen the manuscript, however, we do not believe these additions, while welcomed, are essential for publication of the manuscript as long as the limitations of the work are clearly discussed.  Specifically, we would welcome analysis of "gcy-13::FLP-18 to specifically assess the role of RIM-released FLP-18" (comment of Reviewer #2), and inclusion of "calcium concentrations in body wall muscles that oppose each other in the sinusoidal wave" (comment of Reviewer #3 in relation to Figure 7a).  As noted, inclusion of these data will not be essential for publication but we agree with the reviewers that they would strengthen the manuscript..

[LINK]

Yours sincerely,

Kaveh Ashrafi

Associate Editor

PLOS Genetics

Gregory P. Copenhaver

Editor-in-Chief

PLOS Genetics

Reviewer's Responses to Questions

**Comments to the Authors:**

Reviewer #1: The work of Floorman and Alkema dissects the functional contributions of co-transmission to the behavioural and circuit modules underlying a scape response in C. elegans. The Alkema lab and others have previously shown that the scape response is elicited through the interneurons RIM and AVA and that RIM co-expresses tyramine and flp-18. Tyramine acts on neck muscles to suppress head oscillations and on gabaergic motor neurons to disinhibit body wall muscle. In this work, they examine the role that the release of the neuropeptide flp-18 from the RIM interneurons upon touch stimulation plays in locomotion to elicit scape. They show that the touch-induced scape response consists of a locomotion sequence which includes a reversal, a change in direction through an omega turn and the resumption of forward locomotion with a sustained increase in velocity. Mutants lacking the peptide flp-18 do not display the increase in velocity during forward movement at the end of the scape response and display fewer and abnormally open omega turns. Through epistasis experiments where they test several receptor mutants for their ability to supress the phenotype of flp-18 overexpression, they identify npr-5 as the receptor mediating the effects of flp-18 on locomotion. Based on rescue experiments of npr-5 comparing the effects of driving expression with endogenous versus heterologous promoters, the authors propose that flp-18 action on muscle regulates velocity, whereas action on neurons regulates reversal frequency. Finally, they show that flp-18 increases intracellular Ca2+ levels in body wall muscle in a npr-5- and egl-30/G� q – dependent manner, an effect which (the authors propose) underlies the increase in velocity mediated by the peptide. Together with their previous findings on the regulation of the scape response by tyramine, this work puts forward a model by which co-release of the monoamine and the peptide act synergistically in body wall muscle and antagonistically in neck muscles. Their action on body wall muscles results in an increase in excitability and worm speed whereas their action in neck muscles regulates the suppression and initiation of head oscillations at different steps of the scape sequence.

Co-transmission of neurotransmitters and neuropeptides is a fundamental property of the nervous system which was identified more than 40 years ago (by Thomas Hökfelt and others). Despite its universality, the impact of co-transmission on circuit function and behaviour is till poorly understood. The work of Floorman and Alkema provide a mechanistic logic for how co-transmission can smoothly orchestrate a behavioural sequence and therefore represent an important advance in the field. Furthermore, the parallel regulation of scape behaviour through the co-release of catecholamines and NPY-related peptides in both worm and mammals suggest that a similar logic may be at play.

Mostly, the experiments are carefully done (the behavioural analysis is particularly powerful) and support the conclusions and model proposed by the authors. I only have a few minor comments.

- A link between flp-18 and the increase in forward speed during scape has been established through analysis of flp-18 mutants in Fig. 1. However, the mechanisms underlying this role of flp-18 have not been elucidated. The authors suggest in the discussion that the increase in body bends and Ca2+ levels/excitability in body wall muscles induced by flp-18 may result in higher velocity, but these more likely underly the quality of omega turns. In addition, flp-18 overexpression does not result in an increase in speed. The authors should make note of this in the discussion to clarify the conclusions.

- In relation with the above, data for npr-5 mutants should be included in Figure 1.

- Figure 1 shows that there is a defect in the length of tap-induced reversals in flp-18 mutants but this is not mentioned in the text. This is particularly relevant in light of the findings the authors refer to in the discussion (line 320): “Recent reports have shown FLP-18 acts through NPR-4 and NPR-5 in the AVA and sensory neurons to control reversal lengths”

- The sentence “FLP-18 neuropeptide since it is the only FRFamide known to be expressed in the tyraminergic RIM neurons and the AVA pre-motor interneurons (Kim and Li, 2004; Rogers et al., 2003)” although correct, it is a bit misleading because both RIM and AVA express other FRFamide peptides. Could the authors modify this sentence to clarify this, and also cite the single cell mRNA profiling paper Taylor et al 2021?

- Line 169 “This indicates that FLP-18 release from the escape circuit neurons AVA and RIM is sufficient to enhance body bending and reversals”. The authors should tone down this conclusion since cex-1 drives (ectopic) expression also in AVD, which is part of the reversal circuit.

- The conclusion that npr-5 acts in muscle to enhance body bending and through neurons to increase reversal frequency could be strengthened by driving expression of npr-5 under the panneuronal rab-3 promoter. Also, in this results section referring to figure 6, the action of npr-5 in neurons comes a bit as a surprise because neuronal expression is neither mentioned in the text nor seen in the reporter images. The authors should indicate earlier that npr-5 is also expressed in neurons.

Typo:

- In line 205 “change as a result of tap treatment (Fig. 3f)”. Should say 4f

Reviewer #2: In this manuscript, Florman and Alkema characterize the role of FLP-18, a neuropeptide that RIM co-releases with tyramine during the escape response. First, they show that mutants lacking flp-18 fail to enhance their speed after tap-induced reversals, even though several other aspects of the escape response (e.g. likelihood of reversing) are intact in these mutants. They then show that overexpression of flp-18 increases body curvature, and foraging (nose movements). Correspondingly, flp-18 mutants display fewer omega turns and more incomplete omegas during the escape response. They then use Venus-tagged FLP-18 to show that the FLP-18 peptide is released in response to repeated mechanical stimuli (to repeatedly induce the escape response). An analysis of known flp-18 receptors identifies npr-5 as a receptor whose mutation phenocopies flp-18. In addition, loss of npr-5 suppresses the flp-18 overexpression effects on curvature, head movements, and omega turns. Overexpression of npr-5 under its own promoter causes increased tap-induced curvature. Overexpression in muscle causes a basal increase in curvature. Finally, the authors use GCaMP imaging in muscle to show that FLP-18 overexpression causes increased muscle depolarization, dependent on npr-5 and egl-30.

Overall, this is an interesting paper and the experiments are for the most part convincing, especially the genetic/behavioral analysis. I have a few concerns:

1. The reduction in FLP-18::VENUS signal over hours of repeated tap stimuli is consistent with release of FLP-18 during the escape response, but given that there is no increase in coelomocyte signal, the results seem a bit ambiguous. Could the authors perform this experiment in a unc-31 neuropeptide release-deficient mutant to ensure that the signal change is due to peptide release? Another possible control strain could also be RIM::HisCl to prevent depolarization during tap.

2. The effects of cex-1::FLP-18 expression are only provided in a video format. It would be worthwhile to quantify these results and perform statistical analysis to test whether the results are robust. I would also suggest making and analyzing gcy-13::FLP-18 to specifically assess the role of RIM-released FLP-18. Adding this aspect of cell specificity to the manuscript would strengthen the study.

3. The muscle GCaMP imaging is interesting, but I’m slightly concerned about the quantification. The authors observe increased muscle GCaMP signal in FLP-18 overexpression animals, but there is no control signal to normalize to. Can the authors perform ratiometric imaging (normalizing the GCaMP signal to a co-expressed red fluorophore)? This would rule out the possibility that flp-18 overexpression simply enhances the GCaMP expression levels.

4. Can the authors make a flp-18;npr-5 double mutant to test for additive vs non-additive effects.

Minor concerns:

1. It would be helpful to cite other examples of co-tranmsmission in C. elegans and compare the models arising from these different studies. One example is serotonin and nlp-3 co-release from HSN.

Reviewer #3: In the paper “Co-transmission of neuropeptides and monoamines choreograph the C. elegans escape response”, the authors show that release of FLP-18 during touch-evoked reversals acts through NPR-5 to increase intracellular calcium release in the neck and body muscles. This, in combination with tyramine release from RIM, enhances the overall escape response through increased bending of the head and the body. The authors show that overexpression of FLP-18 results in increased omega turn closure rates and head oscillations, and that this is blocked by npr-5 mutation. Overall, this is an interesting paper that presents a rather complete genetic pathway but a few additional experiments could be added to help support their claims.

Major revisions:

• The order in which data are presented in the figures is at odds with the flow of the main text of the manuscript. Rather than the manuscript walking through each figure, references to each figure occur in many parts of the paper. This makes the manuscript somewhat cumbersome to read, so I would recommend reorganizing the data presentation in the figures to match the logic of the manuscript.

• The authors emphasize in the title as well as the introduction/discussion the novelty of elucidating the various, sometimes opposing, roles of co-transmission of neuropeptide (FLP-18) and monoamine (tyramine). However, the bulk of their data only addresses the contribution of FLP-18 to escape behavior and discussion of tyramine is limited mostly to literature references. Data addressing tyramine can be found in figure 1, where a tdc-1 mutant seems to complete its reversal faster than wild-type (Fig. 1a) but does not show a difference in reversal velocity or forward run velocity (Fig. 1c-d). This is in contrast to the flp-18 mutant, which has decreased velocity in both reversal and forward run. These data would imply that FLP-18 and tyramine release both work in separate pathways, at least in this aspect of the escape behavior. I would like to see some data regarding the involvement of tyramine signaling in their other assays to demonstrate its interaction with FLP-18. For example, does a tdc-1; flp-18 double mutant behave more like tdc-1 or flp-18 single mutant in terms of open omega turns and head movement? Does a tdc-1 mutation affect muscle calcium levels? If they find tdc-1 phenotypes in their assays, can they be rescued by RIM expression?

• The authors mention a few times that the forward run velocity data represents a change in the internal state of the animal, and represents a state of arousal. Their evidence for this is the ~3 minute increase in velocity post-reversal, which they call a “forward run”. However, this specific behavior is not investigated in most of the later experiments with FLP-18 overexpression or npr-5 mutation. I would be interested to see if FLP-18 overexpression increases either the velocity or duration of the forward run.

• Figure 5 explores possible FLP-18 receptors to determine which one is acting in their behavioral pathway. The authors demonstrate that FLP-18 overexpression phenotypes can be blocked by npr-5 mutation. However, none of their data includes single npr-5 mutant behavior without FLP-18 overexpression. This is important to illustrate the nature of their interaction, and how much the npr-5 mutation contributes to the phenotype alone.

Minor revisions:

• The figure references are inconsistently formatted (e.g. 217: “Fig. 3, f”, vs. 225: “Fig. 5d” )

• Lines 114-115 read: “wild-type animals exhibited a markedly elevated forward locomotion rate for approximately 3 minutes (Fig. S1) – a behavior we termed the ‘forward run’”. However, the figure 1 legend, lines 778-779, states: “(d) Mean locomotion velocity during the forward run phase (t = 7-10s)”. Clarify whether “forward run” refers to either a three-minute period or a 7-10s period.

• Figure S2b is missing, and it appears that the figure legend for S2b actually refers to S2a, which is the only panel pictured.

• Lines 226-228 read: “the proportion of open omega turns made by npr-5; FLP-18(+++) animals is significantly increased and similar to the npr-5 single mutants (Fig. 5e)”. While this trend is as stated, the % open omega turns in Fig. 5e by the npr-5; FLP-18(+++) animals is ~65% but in Fig. 3g, the % open omega turns for the npr-5 single mutants is much lower, at around 40%. The WT response remains at around 20% for both graphs.

• In figure 7a, I would be interested in seeing the calcium concentration in body wall muscles that oppose each other in the sinusoidal wave, since one side is contracted and the other, relaxed. With this kind of analysis the authors could see if FLP-18 overexpression increases the calcium concentration but the differential concentration between opposing sides is maintained, or if calcium concentration is increased to the extent that the differential is reduced. The images could also be presented with a colored heatmap representing calcium concentration instead of grayscale to increase the dynamic range of the representative images, which on my screen appear to be saturated in spots.

• In the Molecular Biology and Transgenics section, clarify how the high-expressing vs. low-expressing FLP-18 lines were made. Adding plasmid concentrations to the strain list would help with this.

• Lines 400 – 402 read: “FLP-18 transgenic strains were created by amplifying a 5.8kb sequence of genomic DNA beginning 3.3kb upstream of the open reading frame and containing the flp-18 coding region as well as the 3’ UTR. The extrachromosomal array was stably integrated…”. Please indicate specifically that this region of DNA was cloned into a plasmid and injected. As is it sounds like the amplified PCR fragment was injected. The same applies to the NPR-5 transgenic process.

**Have all data underlying the figures and results presented in the manuscript been provided?**

Reviewer #1: Yes

Reviewer #2: Yes

Reviewer #3: Yes

PLOS authors have the option to publish the peer review history of their article (what does this mean?). If published, this will include your full peer review and any attached files.

Reviewer #1: No

Reviewer #2: No

Reviewer #3: No

---

## [Decision Letter · Decision Letter 1]

11 Feb 2022

Dear Dr %Alkema%,

We are pleased to inform you that your manuscript entitled "Co-transmission of neuropeptides and monoamines choreograph the C. elegans escape response." has been editorially accepted for publication in PLOS Genetics. Congratulations!

Yours sincerely,

Kaveh Ashrafi

Associate Editor

PLOS Genetics

Gregory P. Copenhaver

Editor-in-Chief

PLOS Genetics

Comments from the reviewers (if applicable):

Reviewer's Responses to Questions

**Comments to the Authors:**

Reviewer #1: The revised version of the manuscript satisfactorily addresses all my initial concerns and I think also the ones from the other reviewers.

Reviewer #2: The authors have adequately addressed my concerns.

Reviewer #3: The authors have addressed all of my concerns. Congratulations on an interesting paper

**Have all data underlying the figures and results presented in the manuscript been provided?**

Reviewer #1: Yes

Reviewer #2: Yes

Reviewer #3: Yes

PLOS authors have the option to publish the peer review history of their article (what does this mean?). If published, this will include your full peer review and any attached files.

Reviewer #1: No

Reviewer #2: No

Reviewer #3: No

**Data Deposition**

http://datadryad.org/submit?journalID=pgenetics&manu=PGENETICS-D-21-01371R1

**Press Queries**

---

## [Editor Report · Acceptance letter]

27 Feb 2022

PGENETICS-D-21-01371R1 

Co-transmission of neuropeptides and monoamines choreograph the C. elegans escape response. 

Dear Dr Alkema, 

We are pleased to inform you that your manuscript entitled "Co-transmission of neuropeptides and monoamines choreograph the C. elegans escape response." has been formally accepted for publication in PLOS Genetics! Your manuscript is now with our production department and you will be notified of the publication date in due course.

With kind regards,

Katalin Szabo

PLOS Genetics

On behalf of:
